# An apical ring protein essential for conoid complex assembly and daughter cell formation in *Toxoplasma gondii*

Wei Li [1,2,8] ✉, Oliwia Koczy [3,4,8], Peipei Qin[2], Ignasi Forné [5], Simon Gras[2], Jennifer Grünert[6], Andreas Klingl [6], Simone Mattei [3,7], Elena Jimenez-Ruiz [2] ✉ & Markus Meissner [2] ✉

In *Toxoplasma gondii*, the conoid complex consists of intraconoidal microtubules (ICMTs), preconoidal rings (PCRs), apical polar ring (APR), and the conoid. This organelle plays an important role for initiation of gliding motility, required for host cell invasion and egress. The molecular mechanisms governing stepwise assembly of the conoid complex remain poorly understood. We previously identified CGP, an essential protein required for motility initiation. Here, we demonstrate that CGP is crucial for anchoring FRM1 and other PCR components to mature PCRs, while the initial assembly in daughter cells is unaffected. Cryo-electron tomography of CGP-depleted parasites reveals the absence of the PCRs in the mature parasites, demonstrating that CGP is essential for stabilising the PCRs after replication. Using CGP as bait, we identify a protein required for the early assembly of the nascent conoid complex. The APR scaffold assembly factor (ASAF1) defines the position of the conoid complex before tubulin polymerisation. Depletion of ASAF1 results in failure of conoid complex assembly, disorganised microtubules, and lack of daughter cell formation. Collectively, our findings reveal two essential proteins that play critical roles in the early and late stages of conoid complex formation, providing insight into the mechanisms of conoid complex assembly.

The phylum Apicomplexa comprises over 6000 parasite species that have developed specific adaptations for invasion, replication within, and exit from host cells—processes crucial for their pathogenesis. A defining feature of apicomplexan parasites is the apical complex, which is pivotal throughout the parasite's lytic cycle[1,2]. In *Toxoplasma gondii*, a model Apicomplexa, this includes the conoid complex and unique secretory organelles, micronemes, and rhoptries that contain important factors for motility, host cell invasion, and egress[3]. Interestingly, recent studies highlighted the organisation and anchorage of the rhoptries to the intraconoidal microtubules (ICMTs), two short microtubules inside the conoid during the tachyzoite stage of *T. gondii*[4].

The conoid, formed by a polymer of tubulin, is a motile organelle that can protrude through the apical polar ring (APR) from a retracted

---

[1]Department of Parasitology, College of Veterinary Medicine, Sichuan Agricultural University, Chengdu, China. [2]Experimental Parasitology, Department of Veterinary Sciences, Faculty of Veterinary Medicine, Ludwig-Maximilians-University (LMU), Munich, Germany. [3]European Molecular Biology Laboratory (EMBL), Molecular Systems Biology Unit, Heidelberg, Germany. [4]Candidate for joint PhD degree from EMBL and Heidelberg University, Faculty of Biosciences, Heidelberg, Germany. [5]Protein Analysis Unit, Faculty of Medicine, Biomedical Center (BMC), Ludwig-Maximilians-University (LMU), Munich, Germany. [6]Plant Development, Faculty of Biology, Ludwig-Maximilians-University Munich (LMU), Munich, Germany. [7]European Molecular Biology Laboratory, EMBL Imaging Centre, Heidelberg, Germany. [8]These authors contributed equally: Wei Li, Oliwia Koczy. ✉e-mail: weili93@sicau.edu.cn; elena.jimenez@para.vetmed.uni-muenchen.de; markus.meissner@lmu.de

state upon sensing calcium signals[5,6]. Together with the ICMTs, two ring structures, the preconoidal rings (PCRs) and APR, are formed by multi-protein complexes and are vital for the motility and the correct development of daughter cells. Collectively, here we refer to these as the conoid complex (Fig. 1a)[1,7–10]. From the APR, the minus ends of 22 subpellicular microtubules (MTs) radiate and extend to approximately two-thirds of the parasite's length[10,11]. The PCRs are firmly attached to

the conoid and considered to be the central hub for F-actin polymerisation, crucial for activation of gliding motility required for successful host invasion and egress[1,5,12,13].

According to the current model, the actin nucleator formin 1 (FRM1) localizes at the PCRs, where it polymerizes F-actin and initiates motility in combination with the unconventional myosin, MyoH, localised at the conoid. Together, this generates an apical-basal F-actin

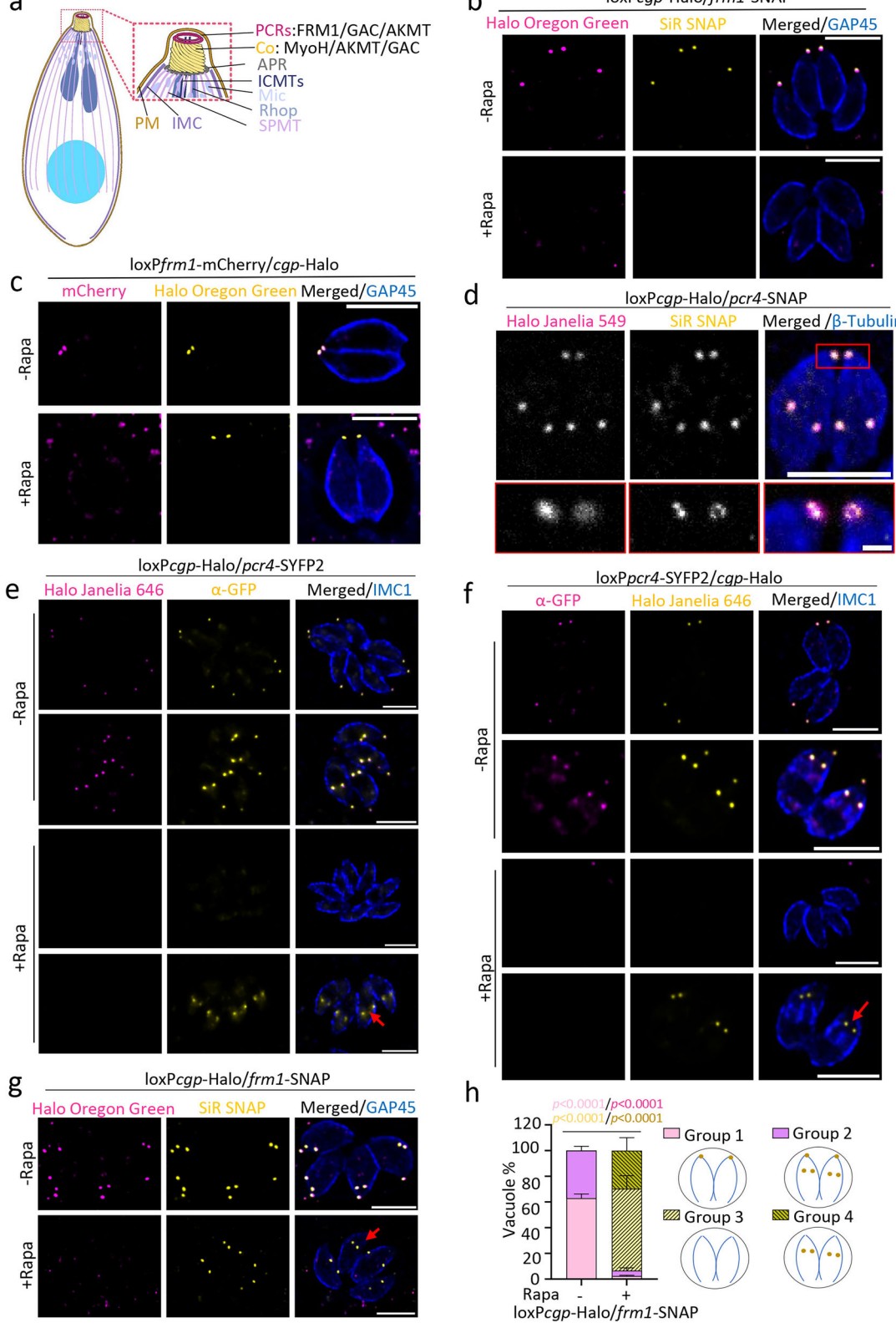

**Fig. 1 | CGP is required for the apical localization of PCR proteins in the mature conoid complex. a** A schematic representation of the apical complex of *T. gondii* with the conoid in a protruded state. APR apical polar ring, Co conoid, ICMTs intraconoidal microtubules, IMC inner membrane complex, Mic microneme, PCRs preconoidal rings, PM plasma membrane, Rhop rhoptry, SPMT subpellicular microtubules. **b** Analysis of CGP deletion on FRM1 which is C-terminally tagged with SNAP (*n* = 3). Deletion of *cgp* was induced with rapamycin (Rapa) and imaging performed 72 h later. Scale bars: 5 μm. **c** Effect of FRM1 depletion on CGP (*n* = 2). Deletion of FRM1 was induced with rapamycin, and imaging performed 72 hours later. Scale bars: 5 μm. **d** Confocal (full parasites images) and STED images (boxed area) show colocalization of CGP with Pcr4 (*n* = 2). Scale bar: 5 μm for confocal images and 0.5 μm for STED imaging. **e–g** Analysis of CGP or Pcr4 deletion on PCR proteins. Marker proteins were C-terminally tagged as indicated. Deletion of *cgp* or *pcr4* was induced with rapamycin and imaging performed 72 h later. **e** Effect of CGP

deletion on Pcr4 (*n* = 2). Pcr4 was still detected in early nascent conoid complex (red arrow) upon CGP depletion. Scale bar: 5 μm. **f** Effect of Pcr4 deletion on CGP (*n* = 2). CGP was still detected in daughter buds (red arrow) upon Pcr4 depletion. Scale bar: 5 μm. **g** Effect of CGP deletion on FRM1 in nascent conoid complex (*n* = 3). FRM1 was present in daughter cells (red arrow). Scale bar: 5 μm. **h** Quantification of FRM1 localisation in loxP*cgp*-Halo/*frm1*-SNAP parasites ± rapamycin. Vacuoles were classified into four groups: (1) FRM1 at the apical tip in non-replicating parasites; (2) FRM1 at nascent and mature conoid complex in replicating parasites; (3) FRM1 absent from mature conoid complex in non-replicating cells; (4) FRM1 absent from mature but present in daughter conoid complex in replicating cells. Schematics show FRM1 (yellow) and GAP45 (blue). Data are mean ± s.d. (*n* = 3). Statistics: two-sided one-way ANOVA with Tukey's test (multiple-comparison adjusted); colour-coded p values indicate conditions.

flow required for parasite motility[5,14–17]. Other proteins critical for gliding motility, such as AKMT (apical complex lysine (K) methyltransferase) and GAC (Glideosome-associated connector), are also accumulated at the apical region[13,18]. Interestingly, the apical localization of GAC requires AKMT activity[18], and both show a dynamic behaviour during motility and invasion[13].

Several proteins have been identified and localised to the PCRs[1,5,12,19], including the structural proteins Pcr4 and Pcr5[5,12]. Interestingly, deletion of Pcr4 or Pcr5 leads to the absence of the PCRs and consequently failure to initiate gliding motility, while daughter cell assembly and biogenesis of the remaining conoid complex (the conoid, ICMTs, the APR) appear unaffected[5].

The stepwise assembly of the conoid complex within the nascent daughter cells has been recently investigated, suggesting that the APR develops prior to the initiation of polymerisation of tubulin structures, which subsequently co-develop with the inner membrane complex[20–22]. Early on, daughter cell construction initiates near the duplicated centrioles. Nascent SPMTs emerge as discrete "rafts," each composed of paired microtubules anchored to the APR, while the conoid and components of the IMC begin to form in parallel. These structures extend toward the centrioles, and eventually, the APR and conoid assemble into a complete ring structure. During this process, it is proposed that SPMTs are nucleated near the centrioles and subsequently incorporated into the forming APR. Ultimately, five distinct SPMT rafts, each containing four to six microtubules, are established, resulting in the nucleation of all 22 SPMTs. The conoid complex, SPMTs, and IMC then extend in a basal direction until the daughter cells mature and bud from the mother cell[20–22]. Given that the conoid and subpellicular microtubules are tubulin-based structures, it has been proposed that the APR serves as microtubule-organizing centre (MTOC) for their formation and assembly[21,22]. Furthermore, recent studies demonstrated a conserved role of γ-tubulin and the γ-tubulin complex in tubulin nucleation[20,23], which appears to be required for polymerisation of spindle microtubules and subpellicular microtubules during division by association with nascent APR.

Previously, through a splitCas9-based phenotypic screen, we identified an apical protein named Conoid Gliding Protein (CGP) (TGGT1_240380) which is required for the initiation of gliding motility[24]. CGP is a large protein (544.89 KDa) containing a CLU-central domain and a tetratricopeptide (TPR)-like domain and is conserved within apicomplexan parasites. The exact mechanism of how CGP contributes to gliding motility remains unclear. Since the observed phenotype upon deletion of CGP closely resembles the one observed for deletion of FRM1, we hypothesised that CGP and FRM1 act in concert to activate F-actin nucleation. Alternatively, CGP could be a structural element of the conoid complex, although deletion of CGP shows no gross effect on apical complex assembly[24].

Our detailed analysis here reveals that CGP is essential for the PCRs' mature structure and is required to anchor FRM1, explaining the defect in gliding motility. Cryo-ET analysis of CGP mutants confirms

the absence of PCRs, suggesting that CGP is necessary for stabilising PCRs during the later stages of daughter cell formation.

Moreover, we identify additional apical components via proximity labelling, including a previously uncharacterised nascent apical polar ring protein, TGGT1_212780. Detailed analysis demonstrates that this cell-cycle specific protein is a key factor for the formation of the nascent conoid complex and daughter cell assembly during endodyogeny.

## Results

### CGP is critical for anchorage of FRM1 to the preconoidal rings

To identify candidate proteins required for actin regulation and host cell egress, we previously used a phenotypic screen based on splitCas9 and performed an initial characterisation of CGP[24]. CGP is a conoid complex protein essential for the initiation of gliding motility and thus plays a critical role in host cell egress. Therefore, we examined the effect of CGP depletion on known proteins required for the initiation of gliding motility, such as the actin nucleator FRM1, GAC, AKMT or MyoH, which are localised at different sub-structures of the parasites' conoid complex[5] (Fig. 1a).

We tagged each of these components in the conditional null mutant for *cgp*, loxP*cgp*-Halo[24]. Upon CGP depletion, no significant changes were observed for the localisation of MyoH or AKMT (Supplementary Fig. 1a, b). However, while the localisation of FRM1 to the PCRs depends on the presence of CGP, the localisation of CGP does not depend on FRM1 (Fig. 1b, c). This suggests that CGP serves as an anchor for FRM1 at the PCRs. Consequently, the defect in gliding motility upon CGP depletion is likely due to the absence of FRM1 at the apical tip, where it may fail to mediate F-actin nucleation, consistent with previous studies[5]. Intriguingly, GAC localises to the apical tip in absence of CGP (Supplementary Fig. 1c) or FRM1 (Supplementary Fig. 1d), suggesting that the initial recruitment of GAC to the apical pole of the parasite occurs independently of F-actin polymerisation at the conoid complex.

### CGP is a critical component of the PCRs

Next, we analysed the localisation of CGP during replication, gliding, and invasion by conducting time-lapse video microscopy. The CGP signal was exclusively detected at the tip of the parasites, suggesting that CGP functions as a fixed structural component rather than playing a dynamic role during these events (Supplementary Video 1).

To define the location of CGP at the apical tip of the parasite in more detail, we performed colocalization studies with the known PCR protein, Pcr4[5], using stimulated emission depletion microscopy (STED), which demonstrated almost perfect co-localisation (Fig. 1d).

Our initial characterisation of CGP demonstrated that it is not required for assembly of some regions of the conoid complex, since RNG2, which localizes to the apical polar ring[7], and SAS6-like (SAS6L), a marker of the conoid[25], were not affected upon deletion of CGP[24]. In contrast, depletion of CGP resulted in the loss of Pcr4 in mature

parasites and vice versa (Fig. 1e, f), supporting the hypothesis that CGP is an important structural component of the PCRs. Interestingly, in the absence of CGP, Pcr4 was still detected at the nascent conoid complex of developing daughter cells, and vice versa (Fig. 1e, f, red arrows), with a similar pattern observed for FRM1 (Fig. 1g, h). This suggests that PCR assembly is initiated during endodyogeny even in the absence of CGP and Pcr4, but both proteins fail to be maintained in mature parasites.

## PCRs are assembled in a two-step process

To test this hypothesis, we performed expansion microscopy (ExM) and analysed the fate of the PCRs during endodyogeny in the presence and absence of CGP using FRM1 as a marker (Fig. 2a). During early endodyogeny, no significant difference in daughter cell assembly could be detected, and the PCRs form a ring-like structure early during daughter bud formation. Importantly, the PCRs seem to stay intact during the early stages of endodyogeny, and later appear to disassemble, leading to complete loss of FRM1 at late stages of endodyogeny (Fig. 2a).

This loss of FRM1 in mature daughters was seen in 100% of all cases and strongly suggests that the PCRs are stable until hatching of the daughter cells, and CGP is required for maintaining the stability of mature PCRs.

The finding that depletion of CGP or Pcr4 did not have an observable effect on the initial assembly of the PCRs in daughter parasites led to the hypothesis that assembly of the PCRs is a two-step process, involving initial assembly in nascent daughter cells, followed by their stabilisation during hatching of daughter cells from the mother. Consequently, in the absence of CGP only the latter step appears to be affected.

## Depletion of CGP leads to complete absence of the PCRs

Since CGP appears to be exclusively required at the PCRs of mature parasites, we wished to analyse the overall structure of the conoid complex in the absence of CGP. Therefore, we performed cryo-electron tomography (cryo-ET) on loxPcgp-Halo parasites. Given the observed knockout efficiency of 75%[24], it was essential to distinguish between induced and non-induced parasites to ensure that only those lacking CGP were analyzed. To achieve this, we developed a protocol enabling fluorescence light microscopy prior to cryo-electron microscopy imaging (cryo-CLEM). While room-temperature confocal microscopy allowed the clear identification of knockout parasites, at liquid nitrogen temperatures, the signal from loxPcgp-Halo labelled with Halo Janelia 646 was too weak to reliably distinguish between wild type (WT) and inducible knockout (iKO) phenotypes. This reduction in fluorescent signal can be attributed to the cryo-confocal setup, which is optimised to minimise heat transfer and prevent devitrification. This setup is incompatible with the immersion lenses typically used at room temperature. Instead, the air objectives used in cryo-light microscopy (cryo-LM) have longer working distances and lower numerical apertures (NA), resulting in diminished signal detection.

To enhance the fluorescent signal and enable clear iKO identification during cryo-LM, we used the loxPcgp-Halo strain with an additional Halo tag on pcr4 (loxPcgp-Halo/pcr4-Halo). In this strain, both Pcr4 and CGP signal simultaneously disappear following induction of the KO. After this optimisation step, parasites were deposited on EM grids and vitrified by plunge-freezing in liquid ethane, before cryo-LM and subsequently cryo-ET on selected parasites. The enhanced signal enabled precise targeting of tilt series data collection for parasites exhibiting either the WT or iKO phenotype (Supplementary Fig. 1e). While analysing the cryo-ET data we noticed that the apical plasma membrane was often disrupted due to the sample preparation procedure and freezing, which is a common issue while vitrifying apicomplexa cells[26,27]. To rule out the possibility of accidental damage or removal of the PCRs during grid preparation, we selected samples with intact apical plasma membranes to assess the presence or absence of

PCRs. Ultrastructural analysis of the reconstructed tomograms revealed that all WT parasites displayed intact PCRs, as shown in previous cryo-ET studies[26,28–30], while parasites lacking CGP completely lacked PCRs (Fig. 2b, Supplementary Videos 2 and 3). This effect was also consistently observed in parasites treated with calcium ionophore A23187 to induce conoid protrusion prior to plunge-freezing (Fig. 2c, Supplementary Fig. 2, Supplementary Video 4). Interestingly, despite the absence of PCRs in the iKO, the conoid was observed to protrude upon calcium ionophore A23187 stimulation in some cases, indicating that even in the absence of efficient apical F-actin formation the conoid is still capable to protrude (Fig. 2c, Supplementary Fig. 2, Supplementary Video 4). These results align with ExM labelling of parasite tubulin, which revealed that conoid protrusion occurred in 5.65% of CGP-depleted parasites upon A23187 stimulation (Supplementary Fig. 3a).

We further assessed whether the cgp knockout affected other organelles or cytoskeletal structures of the parasites. Detailed examination of tomograms from WT phenotype parasites within the iKO cgp sample showed intact micronemes, intraconoidal microtubules, the APR, subpellicular microtubules, conoid, IMC, and several microtubule-associated vesicles (MVs) (Fig. 2c, Supplementary Fig. 2; Supplementary Video 4). However, in 26.09% of cgp iKO parasites (6 out of 23 analysed), rhoptries could not be observed within the tomogram field of view. Additionally, 21.74% (5 out of 23) of the CGP-depleted parasites lacked the apical vesicle (AV), a component of the rhoptry secretion machinery[31], whereas this phenotype was not observed in WT parasites (Fig. 2c and Supplementary Fig. 2; Supplementary Video 4). Furthermore, CGP depletion resulted in the absence of the apical accumulation of non-discharge protein Nd6, an AV marker[31], in 18.03% of the parasites, and a reduction in rhoptry secretion, as indicated by the decreased percentages of phosphorylated STAT6-positive host cell nuclei (Supplementary Fig. 3b, c).

In summary, CGP is a crucial component for the stability of the PCRs, exclusively in mature parasites. Additionally, our findings highlight the potential importance of PCRs in ensuring the proper attachment of rhoptries to the apical complex.

## Comparative proximity labelling identifies apical proteins

Our findings demonstrate that nascent PCRs are assembled in a stepwise manner, with CGP being exclusively required for the stability of mature PCRs, while early assembly appears to be initiated normally during endodyogeny, leading to the hypothesis that other, yet unknown conoid complex proteins are required for the early assembly of the conoid complex. To this end, we endogenously tagged CGP and FRM1 at their C-termini with TurboID and performed comparative proximity labelling[32] on intracellular parasites. Biotin labelling was performed for 6 h and successful biotinylation of the conoid complex was confirmed via immunofluorescence analysis (IFA) and Western blotting (Supplementary Fig. 4a, b).

Analysis of the comparative datasets revealed a similar pool of enriched proteins compared to wild-type tachyzoites (Supplementary Fig. 4c, d, Supplementary Table 1). Notably, CGP and FRM1 were significantly enriched in each other's dataset, further confirming their presence on the same structure. To narrow down the list of candidate proteins, we applied the following criteria: (1) significant enrichment (fold change over 4.5); (2) a phenotypic score below −1, indicating an important function during the asexual cycle of the parasite[33] or previous reports to be essential; and (3) either known localization at the conoid complex based on published localization studies, or predicted apical localization with a final probability score of 1 or no assigned localization or any predicted localization (regardless of apical or not) with a final probability score below 1, predicted by HyperLOPIT[34]. This process resulted in a list of 15 candidates for further investigation (Supplementary Table 1) of which we successfully tagged 13 (Supplementary Fig. 5a–c).

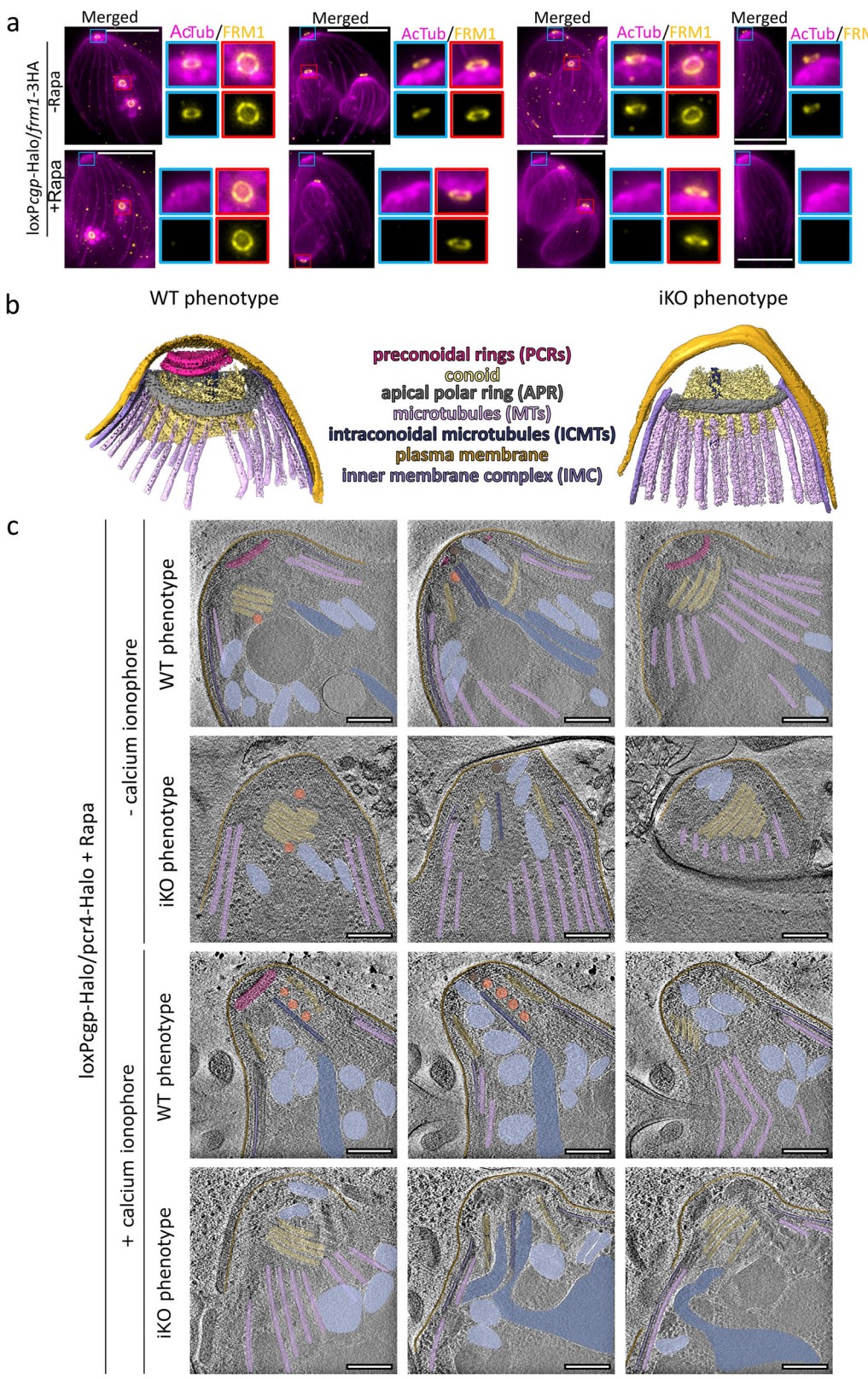

Several previously described proteins, such as GAC, AAMT (apical annuli methyltransferase), and AKMT, were identified using this approach[13,18,35]. However, the depletion of either CGP or FRM1 did not alter their apical localization (Supplementary Fig. 1b–d, 6a–c). Although AKMT is required for the recruitment of GAC to the apical pole of the parasite[18], its deletion did not affect the localization of CGP and FRM1 (Supplementary Fig. 6d, e).

Three PCR proteins, ICAP16 (indispensable conserved apicomplexan protein 16), TGGT1_284620 and TGGT1_292170, were identified as enriched in both TurboID datasets[25,36]. ICAP16 is known to mildly affect host invasion[36]. Its localisation did not depend on FRM1 (Supplementary Fig. 6f), but appears to require CGP for its recruitment to the PCRs, since the loss of CGP resulted in over 40% of vacuoles missing the ICAP16 signal in mature parasites (Supplementary Fig. 6g, h).

**Fig. 2 | Preconoidal rings are missing in CGP-depleted parasites. a** Expansion microscopy of conditional *cgp* knockout (KO) parasites at various stages of endodyogeny, co-stained with acetylated alpha-tubulin (AcTub) and FRM1. In the absence of inducer, FRM1 localizes to the apical complex throughout replication and remains associated with the mature conoid after the parasites emerge from the mother cell. Upon CGP depletion, FRM1 forms a ring with normal appearance during early and mid-stages of endodyogeny. However, as parasites near completion of budding and prepare to exit the mother cell, FRM1 signal is lost in mature parasites. Blue insets: Apical complex of the mother cell—FRM1 is consistently absent from the mature conoid. Red insets: Apical complex of developing daughter cells—FRM1 localizes to nascent conoid complex until the parasites exit the mother (n = 2). Scale bars: 10 µm. **b** Segmentation of reconstructed tomograms of WT and iKO parasites with focus on the apical complex. Dark pink – PCRs, yellow – conoid, grey –APR, light pink – microtubules, dark blue – ICMTs, orange – plasma membrane, violet - IMC. **c** Slices of reconstructed tomograms of apical complexes of iKO *cgp* parasites targeted with cryoCLEM *(n = 2)*. Colors highlight different structures matching panel b. Additional structures are highlighted in light blue – micronemes, blue – rhoptries, orange – microtubule-associated vesicles, brown – apical vesicles. See Supplementary videos 2, 3 and 4 for full tomograms. Scale bar 200 nm.

TGGT1_284620, a hypothetical protein now termed dispensable apical protein 1 (Dap1), was previously detected as a potential Fission 1 (Fis1) interactor via yeast two-hybrid interaction screen, but not detected via immunoprecipitation assay[37]. Dap1 failed to localize to the apical tip in mature cells in the absence of CGP, although FRM1 was not required for its apical localization (Supplementary Fig. 7a, b). However, Dap1 was dispensable for the localization of CGP and other apical proteins (Supplementary Fig. 7c–g), which is consistent with the observation that parasite propagation was not impacted when Dap1 was deleted (Supplementary Fig. 7h).

Lastly, TGGT1_292170 is annotated as a putative histone lysine methyltransferase, which we have now named preconoidal lysine methyltransferase (PCKMT). Although a deeper characterisation of this essential protein is being carried out in a separated study (Qin et al., in preparation), we demonstrate here the localisation of PCKMT at the PCRs where it shows colocalization with Pcr4 (Supplementary Fig. 8a). As expected, PCKMT was absent in mature PCRs upon CGP depletion (Supplementary Fig. 8b). However, as it occurs with Pcr4, FRM1 and Dap1, PCKMT remains present in budding daughter cells upon depletion of CGP (Fig.1e, g; Supplementary Fig. 7a and Supplementary Fig. 8c, red arrows).

To assess the expression levels of apical proteins that lose their apical localization upon CGP depletion, we performed Western blot analysis. This revealed a reduction in Pcr4 and ICAP16 levels following induction of the *cgp* knockout (Supplementary Fig. 8d), suggesting that these proteins are degraded in mature PCRs. While we attempted to quantify FRM1, PCKMT, and Dap1 under the same conditions, the results were inconclusive—likely due to their low abundance and large molecular size. Nonetheless, we cannot exclude the possibility that their total levels remain unchanged or are modestly reduced in the absence of CGP.

Collectively, these proteins appear to play no role in PCR assembly or stability. Furthermore, PCKMT and Dap1 localise exclusively at the PCRs, since deletion of CGP results in their absence from the apical complex. In contrast, the association of ICAP16 depends only partially on the presence of CGP.

## Identification of an early daughter-specific apical protein: ASAF1

Among the 15 selected candidates, we found three proteins predicted to be expressed in a cell cycle-dependent manner, with RNA expression levels peaking during mitosis (Supplementary Fig. 9a)[38]. Localisation analysis demonstrated that TGGT1_238170, known as IMC-associated protein 3 (IAP3), has indeed cell cycle-dependent expression and displayed dynamic localization patterns during replication (Supplementary Fig. 9b), consistent with a recent study[39]. However, since IAP3 did not co-localise with CGP in the nascent daughter conoid complex, it was excluded from further analysis.

In contrast, TGGT1_210430 was expressed throughout the cell cycle and showed partial co-localisation with apical markers in developing daughter cells (Supplementary Fig. 9c), suggesting a possible role in the assembly of the nascent conoid complex. We generated an inducible knockout mutant using the DiCre system[40] (Supplementary Fig. 9d). However, plaque assays revealed that TGGT1_210430 is dispensable for parasite propagation (Supplementary Fig. 9d, e). This

finding suggests that its role in daughter conoid complex formation is likely non-essential under standard conditions. Consequently, it was not pursued further in this study.

Finally, TGGT1_212780 is a large protein (253 kDa) conserved in coccidian parasites and is predicted to contain coiled-coil regions but lacks other identifiable domains. It is exclusively expressed during parasite replication, where it colocalises with CGP in forming daughter cells, but is undetectable at the PCRs of mature parasites (Fig. 3a, Supplementary Fig. 9f). Based on these findings, we focused our analysis on TGGT1_212780, which we named APR scaffold assembly factor (ASAF1).

## The early recruitment and stepwise development of ASAF1 at the APR is associated with microtubules formation

During endodyogeny, the centrosome is the first organelle to duplicate, followed by the formation of the nascent conoid complex and the construction of daughter cells. We found that ASAF1 can be detected after the duplication of centrin 1 (Supplementary Fig. 10a), a marker of the centrosome, but prior to the appearance of markers for the conoid complex at the daughter cells, such as MyoH and RNG2 or initiation of daughter bud formation (Supplementary Fig. 10b, c), making ASAF1 the earliest known marker to be recruited to the nascent conoid complex.

We analysed the location of ASAF1 relative to other markers of the conoid complex during endodyogeny using STED microscopy, which showed that ASAF1 is positioned below CGP, encircles MyoH, and colocalizes with RNG2, indicating that ASAF1 is an apical polar ring protein (Fig. 3a). Interestingly, in early daughter cells, ASAF1 initially forms an arc or incomplete ring that partially encloses CGP (Fig. 3a, red arrows). Later, this ring closes (Fig. 3a, blue arrows) surrounding the rest of apical structures.

To further characterize ASAF1 dynamics during replication, we performed expansion microscopy (ExM), which enlarged the parasites approximately fourfold. At the onset of mitosis, centriole duplication and spindle microtubule formation occur, and ASAF1 appears as a small, elongated structure (Fig. 3b, panel 1; Supplementary Video 5a). During early budding, ASAF1 adopts an arc-like shape and associates with nascent daughter cells, where the emerging microtubules are arranged into several rafts. Even before the conoid and MT rafts are fully formed, ASAF1 occasionally appears as a closed ring (Fig. 3b, panel 2, red arrow; Supplementary Video 5b). As mitosis progresses into anaphase, flower-like assemblies mature into dome-shaped structures, subpellicular MTs organize into five distinct rafts, and a complete ASAF1 ring becomes clearly visible (Fig. 3b, panels 3 and 4; Supplementary Videos 5c, d). Finally, once subpellicular MTs become evenly distributed in the daughter cells and only the spindle poles remain visible (with spindle MTs absent), ASAF1 is no longer detectable (Fig. 3b, panel 5; Supplementary Video 5e).

The development of nascent tubulin correlates well with recent research[20,22]. The early co-occurrence of ASAF1 with tubulin, despite its recruitment before MTs development, suggests that ASAF1 acts as a platform for the formation of the APR and consequently for recruitment of additional APR proteins, including the γ-tubulin, necessary for the initiation and growth of nascent MTs[20,23].

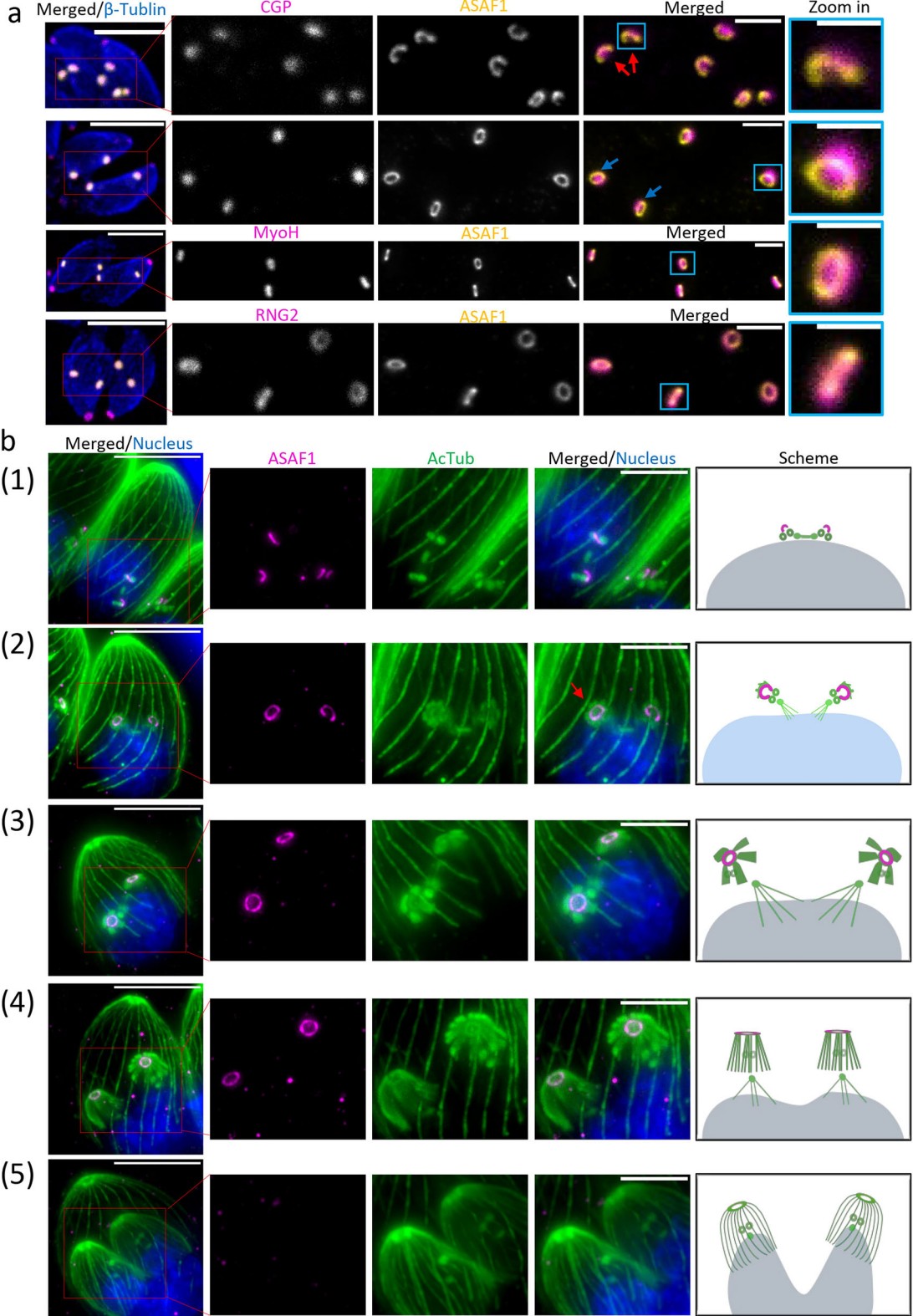

## ASAF1 is crucial for daughter cell formation

To investigate the function of ASAF1, we generated a conditional knockout strain in which *asaf1* is floxed and tagged with a SYFP2 tag at its C-terminus in the RHDiCreΔKu80 strain, named loxP*asaf1*-SYFP2. Excision of the *asaf1* gene resulted in no plaque formation (Supplementary Fig. 10d-e), highlighting its essential role in parasite growth. Subsequent analyses by standard (non-expanded) IFA imaging and ExM revealed severe alterations in parasite morphology, as evidenced by GAP45, IMC1, and microtubules staining (Fig. 4a, b; Supplementary Fig. 10f–h). Interestingly, nuclear division was not significantly affected, and formation of spindle microtubules appeared normal (Fig. 4b; Supplementary Video 6). In contrast, no typical daughter IMC formation could be observed in the absence of ASAF1 (Fig. 4a).

**Fig. 3 | ASAF1 is a daughter cell specific APR protein formed in a stepwise manner. a** Colocalization analysis of ASAF1 with indicated apical markers during endodyogeny (*n* = 2). ASAF1 colocalizes well with APR marker RNG2 but not with PCRs marker CGP or conoid marker MyoH. Red arrows denote ASAF1 appearing as an arc or incomplete ring in the early stages. Blue arrows denote ASAF1 appearing as a complete ring. Merged images are confocal images, and the scale bar is 5 μm. Inset images are STED imaging pictures of the boxed areas, and the scale bar is 1 μm. The scale bar for the blue inset is 0.5 μm. **b** ExM images show localisation of ASAF1 at different stages of endodyogeny. Co-staining was performed using acetylated alpha tubulin antibody (*n* = 3). (1) At the onset of replication, ASAF1 is detected as a small arc after spindle microtubule formation. Note that the formation of other tubulin-based structures (subpellicular microtubules) has not yet been initiated. (2) ASAF1 appears as a major arc and associates with nascent subpellicular microtubules. The red arrow indicates that even before the nucleation of all SPMTs, ASAF1 is already detected as a complete ring. (3) With the formation of subpellicular microtubules, ASAF1 forms a complete ring. Flower-like tubulin structures are observed with subpellicular microtubules organized into five distinct rafts. (4) During late stages of endodyogeny, ASAF1 is present as a complete ring, and subpellicular microtubules are transformed into the typical dome-like structures. (5) At late stages, when daughter buds are well organised, microtubules show the typical cage with even distribution, ASAF1 is no longer detectable. Images correspond to Supplementary video 5. Scale bar: 5 μm.

Interestingly, the organization of subpellicular MTs was severely disrupted, with some vacuoles displaying dot-like structures that, upon closer examination by ExM, are likely to be spindle MTs and centrioles (Fig. 4b; Supplementary Fig. 10h; Supplementary Video 6, white arrowhead). The observation of MTs anterior to spindle MTs in the absence of ASAF1 suggests that formation of MTs still occurs in a chaotic and disorganised way. This is in good agreement with a role of ASAF1 for the formation of the APR that serves as MTOCs for daughter bud formation (Fig. 4b).

This finding was further confirmed using transmission electron microscopy. In the WT samples, 100% of vacuoles exhibited a typical structure with correctly formed tachyzoites. Among these, 20–30% showed visible daughter cell formation, and approximately one-third of those allowed clear visualization of the daughter cell conoids. In contrast, parasites depleted of ASAF1 still showed a relatively normal nuclear organisation during division, but in no instance, we were able to identify any daughter conoid complex or correct IMC formation (Fig. 4c, Supplementary Fig. 11). In some cases, formation of membrane folds was observed in *asaf1* knockout parasites that seem to originate from the mature parasites (Fig. 4c, Supplementary Fig. 11, white arrows).

Next, we performed time-lapse video microscopy to analyse the behaviour of daughter cell formation in absence of ASAF1. IMC1 was endogenously tagged with mCherry in the loxP*asaf1*-SYFP2 parasites. As expected, in non-induced knockout parasites, the budding of parasites from the maternal cell resulted in a doubling of parasite numbers after each replication cycle (Supplementary Fig. 12, Supplementary Video 7). In contrast, upon induction of the knockout, despite the enlargement of the parasites, no daughter IMC formation could be observed (Supplementary Fig. 12, Supplementary Video 7).

To study the behavior of tubulin, we transiently expressed a mCherry-tagged tubulin copy in the loxP*asaf1*-SYFP2 parasites. In controls, we observed the early formation and extension of daughter buds; once fully developed, the daughter cells emerged from the mother cell, and the maternal tubulin disappeared (Supplementary Fig. 12, Supplementary Video 7), indicating the completion of a replication cycle. In the knockout-induced parasites, only tubulin dot-like signals corresponding to spindle microtubules were observed, with no detectable formation of daughter cell subpellicular microtubules (Supplementary Fig. 12, red arrows, Supplementary Video 7).

Taken together, our results indicate that ASAF1 is required for the early assembly of the APR and, therefore the organised formation of subpellicular microtubules, which provide the scaffold necessary for daughter cell construction.

### ASAF1 is not involved in centrosome duplication but required for Striated fibre assemblin and γ-tubulin organisation at the conoid complex

Having established that ASAF1 is critical for formation of the APR and consequently the organised formation of the subpellicular MTs, we wished to analyse if processes acting upstream, such as the duplication of the centrosome, centrocone, and kinetochore are affected in the absence of ASAF1. Therefore, we examined the fate of centrin1, chromodomain containing protein (Chromo1), and Nuf2, which are markers for the centrosome, centromere, and kinetochore, respectively[41–43] and found that duplication is unaffected (Supplementary Fig. 13a–c).

Striated fibre assemblin 2 (SFA2), a component of the striated rootlet fibre, is critical for daughter bud formation, connecting the centrosome to the apical structures in developing daughter cells[20,44]. In ASAF1-depleted parasites, fibres were still formed and appeared to associate with centrioles and tubulin structures during early mitosis, similar to control parasites. However, upon completion of replication, SFA2 disappeared from the conoid complex region in ASAF1-depleted parasites, confirming its association with the conoid complex (Fig. 5a).

We further investigated the fate of γ-tubulin, which is suggested to be responsible for the nucleation of tubulin-based structures and the correct SPMT formation[20,23], upon ASAF1 depletion. In both WT and ASAF1-depleted parasites, γ-tubulin was detected at the pole of spindle microtubules and was also associated with centrioles during early mitosis. During late mitosis, it remained restricted to the spindle pole. However, the horseshoe-like pattern of γ-tubulin associated with the conoid, observed in WT parasites during early mitosis, was not detected in ASAF1-depleted parasites (Fig. 5b).

### ASAF1 is required for stabilising the conoid complex substructures during their assembly

Since IFA and ultrastructural images suggest defects in conoid complex (conoid and the APR) formation, we investigated the effect of ASAF1 deletion on proteins localised to the different compartments of the conoid complex, PCRs, conoid, and APR using the established markers FRM1/CGP, MyoH, and RNG2, respectively.

First, we found that depending on the marker, multiple signals could be observed in 38 – 86% of the vacuoles in absence of ASAF1 (Supplementary Fig. 13d–i). These protein signals form randomly localised clusters within the parasite (Fig. 6a–c, Supplementary Fig. 13d–i). Next, we wished to investigate if these clusters co-localised or if markers for individual sub-structures of the conoid complex lost their association usually found in developing daughter parasites. No co-localization of the PCR marker CGP with the conoid (MyoH) or the APR (RNG2) was detected (Fig. 6a–c). However, CGP still co-localized with another PCR resident, FRM1 (Fig. 6c). Although Western blot analysis for CGP, FRM1, and RNG2 was unsuccessful, MyoH exhibited a decreased expression level (Supplementary Fig. 13j). These results indicate that ASAF1 is required for the assembly and initial stability of the conoid complex until it is fully formed and stable (Fig. 7), potentially functioning like a scaffold for the ordered assembly of individual conoid complex substructures.

## Discussion
### CGP is a key protein required for stability of mature PCRs
The motility of *T. gondii* is initiated by proteins localised at the conoid complex, where critical proteins, including the apical actin nucleator FRM1, MyoH, and GAC are located[5]. Previously, we employed a phenotypic screen to identify proteins involved in actin regulation and identified a high molecular weight hypothetical protein of 544.89 KDa, CGP, that is conserved in apicomplexan parasites and localises to the conoid

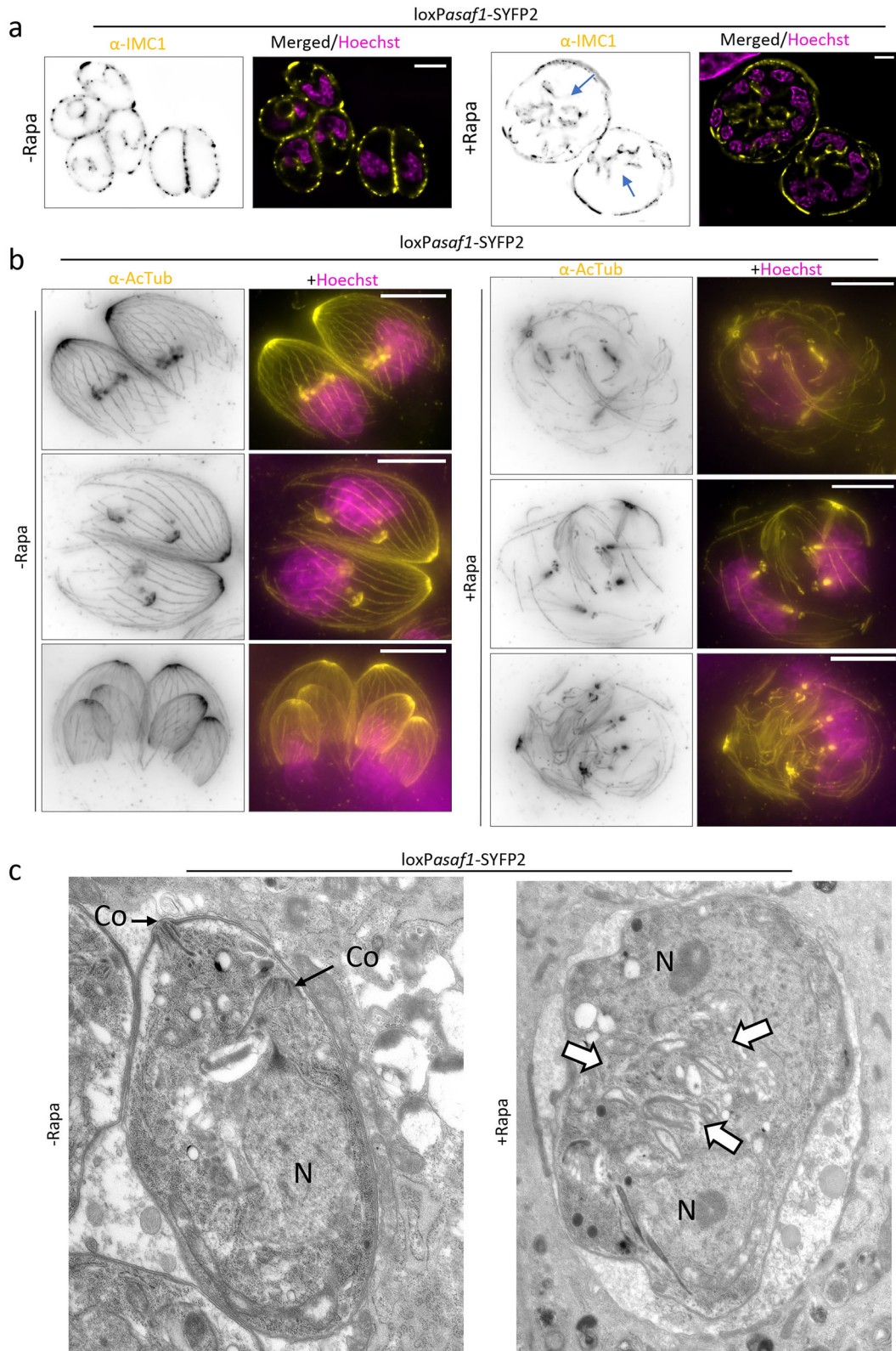

complex of the parasite[24]. Since deletion of CGP results in a similar phenotype as observed for other gliding initiation factors, we hypothesized that it is linked to the apical regulation of F-actin dynamics by being functionally linked to the apical actin nucleator FRM1.

Our data demonstrates that FRM1 is absent from the apical tip in mature conoid complex upon deletion of *cgp*, which may lead to the absence of F-actin at the PCRs and, consequently, the loss of apical-

basal F-actin flow required for initiation of motility[5,14,15,17,24]. Interestingly, despite potential abrogation of apical F-actin formation, the actin binding protein GAC[18] localizes to the apical tip of the parasite, suggesting that GAC may interact with other proteins at the apical complex to maintain its position until motility is triggered and F-actin binding activated. Instead, this actin-independent association appears to be dependent on the methyltransferase AKMT, as previously

**Fig. 4 | Depletion of ASAF1 leads to a defect in daughter cell formation. a** ExM images demonstrate the effect of ASAF1 depletion on parasite replication. loxP*asaf1*-SYFP2 parasites were grown in presence or absence of inducer for 48 h, before fixation. Parasites were labelled with anti-IMC1 to analyse replication (*n* = 2). Depletion of ASAF1 results in no discernible daughter cell formation, although remnants of potentially nascent IMC (blue arrows) can be detected in the cytosol of the parasites. Interestingly, nuclear division (stained in magenta) still occurred. Only one stack is shown for IMC1. Scale bar: 10 μm. **b** ExM images show the effect of ASAF1 depletion on tubulin-based structures (*n* = 2). Depletion of ASAF1 caused disorganized nascent subpellicular microtubules that in many cases appear to be still attached to the centrosome. No daughter cell formation was observed. Images correspond to Supplementary Video 6. Scale bar: 10 μm. **c** TEM images of LoxP*asaf1* non-induced and induced parasites. KO-induced parasites exhibit severe morphological defects, and no daughter cell formation was observed. White arrows indicate membrane-folded structures, probably originated from the excess of membrane generated in the mature cell during cell division. Although more than 20 parasites (*n* > 20) were analysed, no conoid-like structures could be detected. Co Conoid, N Nuclei. Scale bar: 1000 nm.

shown[18]. Identifying the targets of this methyltransferase could provide insight into the recruitment of GAC to the conoid complex.

Since the location of FRM1 was described to depend on intact preconoidal rings[5], we performed cryo-ET using a recently developed protocol (cryo-CLEM) that allows us to identify and correlate KO-mutants. During these experiments, we realised that the apical plasma membrane is often disrupted and therefore to avoid imaging of artefacts, we only analysed parasites with a fully intact apical membrane. Analysis of *cgp* iKO parasites conclusively demonstrated the absence of the preconoidal rings at the mature conoid complex. Interestingly, we observed protrusion of the conoid in absence of PCRs, although in only small percentages of parasites, suggesting that the conoid can be protruded at least to a certain extent independently of FRM1.

The absence of docked rhoptries in some CGP-depleted parasites in our tomogram suggest that CGP might play a role in stabilizing apical complex structures involved in rhoptry positioning, such as AV. However, we never observed a parasite lacking both AVs and rhoptries, indicating that AV loss alone does not account for the absence of rhoptries in our tomograms. One possible explanation is technical: the thin and elongated rhoptry necks may be poorly resolved in thicker tomographic samples that were not subjected to cryo-FIB milling. Supporting this, we observed two instances in wild-type parasites where docked rhoptries were not visible, suggesting this may be a general limitation of the method. In the future, it may be interesting to employ other high-resolution imaging techniques, such as focused ion beam scanning electron microscopy (FIB-SEM), to accurately assess rhoptry formation and apical positioning. This could help address whether PCRs play a role in rhoptry biogenesis or anchoring rhoptries. If the absence of PCR also has a significant effect on rhoptry positioning at the apical tip, similar to defects seen after ICMAP deletion[4], which affect the ICMTs, this may suggest a link between the ICMTs and PCRs. However, we were unable to observe a direct effect on ICMTs in absence of the PCRs, confirming that ICMT formation doesn't depend on PCRs.

**Absence of CGP does not affect initial assembly of nascent conoid complex during replication**
The absence of CGP and other previously identified structural proteins of the PCR, such as Pcr4[5] did not affect the assembly of the nascent PCRs during replication. Surprisingly, even the association of PCR proteins, like FRM1, Dap1, or PCKMT is not affected during conoid formation in replicating parasites. Instead, FRM1 is present during early, mid, and late budding stages until daughter cells hatch from the mother. This suggests that CGP is exclusively required for maintaining the stability of the PCRs on mature parasites, but not during their formation.

Based on these data, we propose a two-step process for PCR formation: initial assembly requires factors that recruit PCR proteins to the nascent conoid complex, independently of known structural proteins. Later, these interactions are stabilised, once the PCRs are fully developed and hatch from the mother cell (Fig. 7). This final maturation and stabilisation of PCRs requires CGP, Pcr4, and Pcr5.

**Identification of ASAF1, an essential APR assembly factor**
To identify potential assembly factors, we performed comparative proximity labelling using CGP and FRM1 as bait and identified known and previously uncharacterised PCR proteins. We successfully identified several PCR proteins, including PCKMT, Dap1, and ICAP16, which are not required for conoid formation and await further characterisation (Qin et al. in preparation). In contrast, the early APR protein, ASAF1, is critical for daughter cell assembly. Closer examination of the phenotype demonstrated that depletion of ASAF1 disrupts organised polymerisation of nascent subpellicular MTs and assembly of the conoid complex.

**ASAF1 is exclusively required for the formation of daughter conoid complex**
Recent studies highlight a temporal and spatial hierarchy of conoid complex formation in relation to centriole duplication and nuclear division[20–22]. Based on its conservation in most eukaryotes, a role of γ-tubulin in organisation of MTs in apicomplexans has long been expected and recently shown independently to be indeed required for spindle pole formation and polymerisation of the subpellicular MTs. Interestingly, the study by the Gubbels lab suggests that the daughter scaffold is initiated close to the centrioles and relies on γ-tubulin and the SFA fibre. The subpellicular MTs are initially nucleated close to the centrioles and subsequently added to the forming APR, which leads to the organisation of the subpellicular MTs. According to this model, initiation of nascent, subpellicular MTs occurs close to the centriole, before being transferred to the APR, which then organises the formation of the typical subpellicular basket[20,23].

In this study, we found that upon ASAF1 depletion, γ-tubulin is lost from the conoid complex region but remains associated with centrioles and the spindle pole. This explains the normal formation of spindle MTs and the occurrence of nuclear division. Notably, we also observed a population of long, unorganised MTs anterior to the nucleus, which, in good agreement with the model mentioned above, appear to radiate from centrioles into the cytosol of the parasite, and can be explained by the absence of APR that anchors them to form the typical basket.

Consistent with this hypothesis, ASAF1 can be detected in daughter cells after centriole duplication, but before MT formation. Furthermore, striated fibre protein SFA2, which is required for establishing the daughter cell scaffold, shows no major alterations upon deletion of ASAF1. The SFA fibre arises from between the centrioles and connects to the nascent apical tip of the parasite in control parasites. In the case of ASAF1 depletion, SFA2 can still be seen connected to the centrioles in absence of the nascent conoid complex, suggesting that ASAF1 exclusively functions in conoid complex assembly. Consistent with this, critical structures for nuclear division, such as the centrocone, centrosome, and kinetochore, remain unaffected.

In the absence of ASAF1, components of different conoid complex sub-structures can still be detected in most cases. Interestingly, these sub-structures (APR, conoid, and PCRs) do not co-localise but are randomly distributed within the cytosol of the mother cell, indicating that the ASAF1-ring acts as an assembly point for the recruitment of conoid complex proteins.

ASAF1 depletion also results in the absence of typical IMC formation in replicating cells. Instead, remnants of potentially nascent IMC structures (Fig. 4a, c) can be observed in the cytosol of the parasites. This indicates that initiation of IMC biogenesis still occurs;

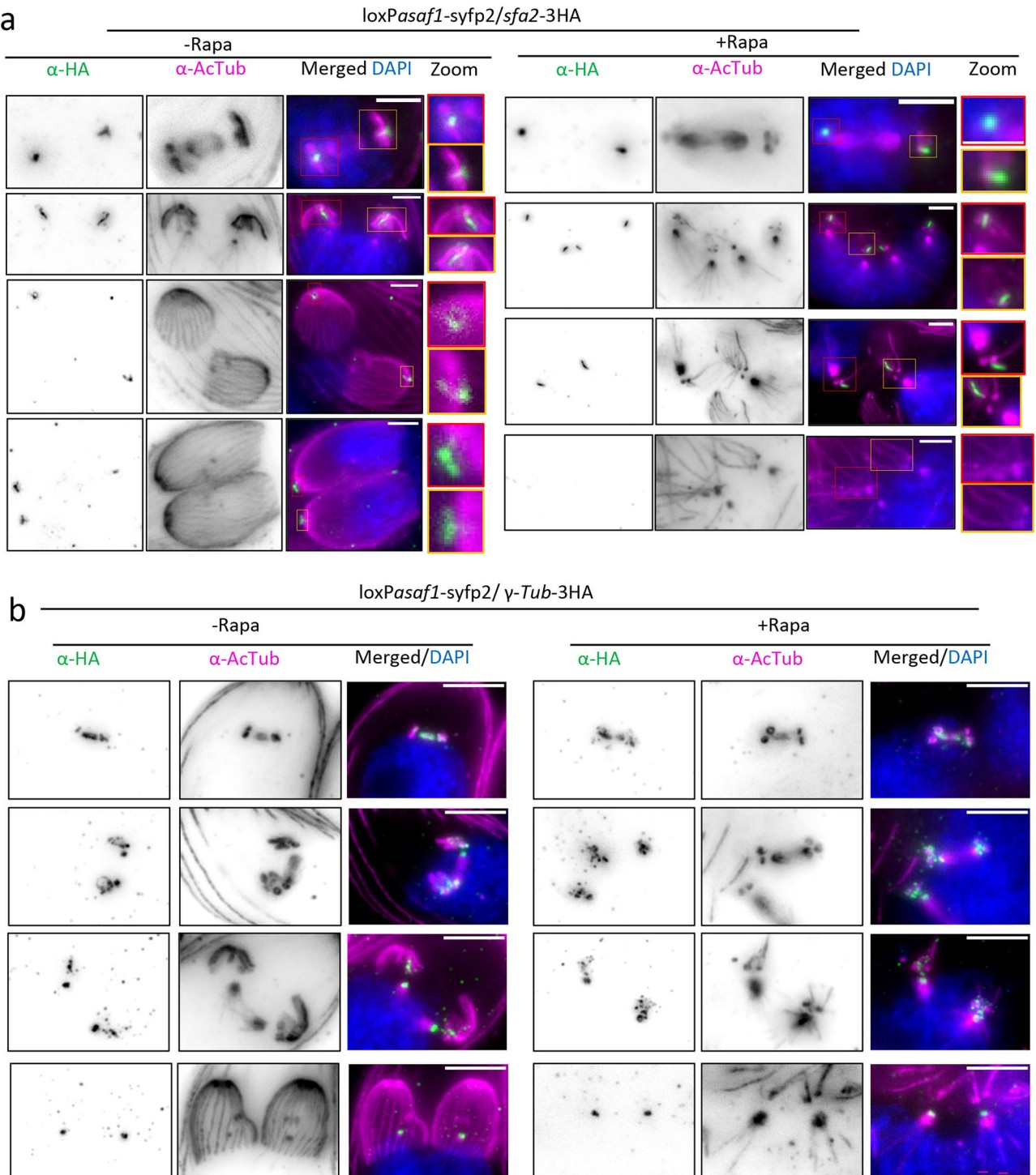

**Fig. 5 | ASAF1 depletion disrupts the localization of SFA2 and γ-tubulin during endodyogeny. a** ExM images show the effect of ASAF1 depletion on the striated fibre component SFA2, which plays a pivotal role in robust spatial and temporal organisation of parasite division (*n* = 2). In WT parasites, SFA2 connects the apical structures and centrioles in early endodyogeny, with the SFA2 signal retained only in the apical region during late endodyogeny. Upon ASAF1 depletion, SFA2 is observed in early replicating cells, associating with centrioles and microtubules, indistinguishable from the situation in wt. However, later it is not detected due to the absence of the conoid. Scale bar: 3 μm. **b** ExM images show the effect of ASAF1 depletion on γ-tubulin (*n* = 2). Upon ASAF1 depletion, γ-tubulin is no longer detected at the conoidal region but remains present at the pole of spindle MTs and around the centrioles. Scale bar: 5 μm.

however, in the absence of apical polarity and the characteristic SPMT basket, which normally supports alveolar plate architecture[20], regular IMC plates fail to form.

Based on these findings, we propose the following model for formation and maturation of the conoid complex (Fig. 7). Shortly after initiation of replication and duplication of the centrioles, ASAF1 forms an incomplete ring that acts as a recruitment point for the individual components of the conoid complex. We suggest that it acts to anchor and stabilise the individual sub-structures of the conoid complex, possibly by interacting with SFA2. As suggested previously, once APR is formed, nascent subpellicular microtubules are anchored to the APR. Once daughter cells are completed, the conoid complex

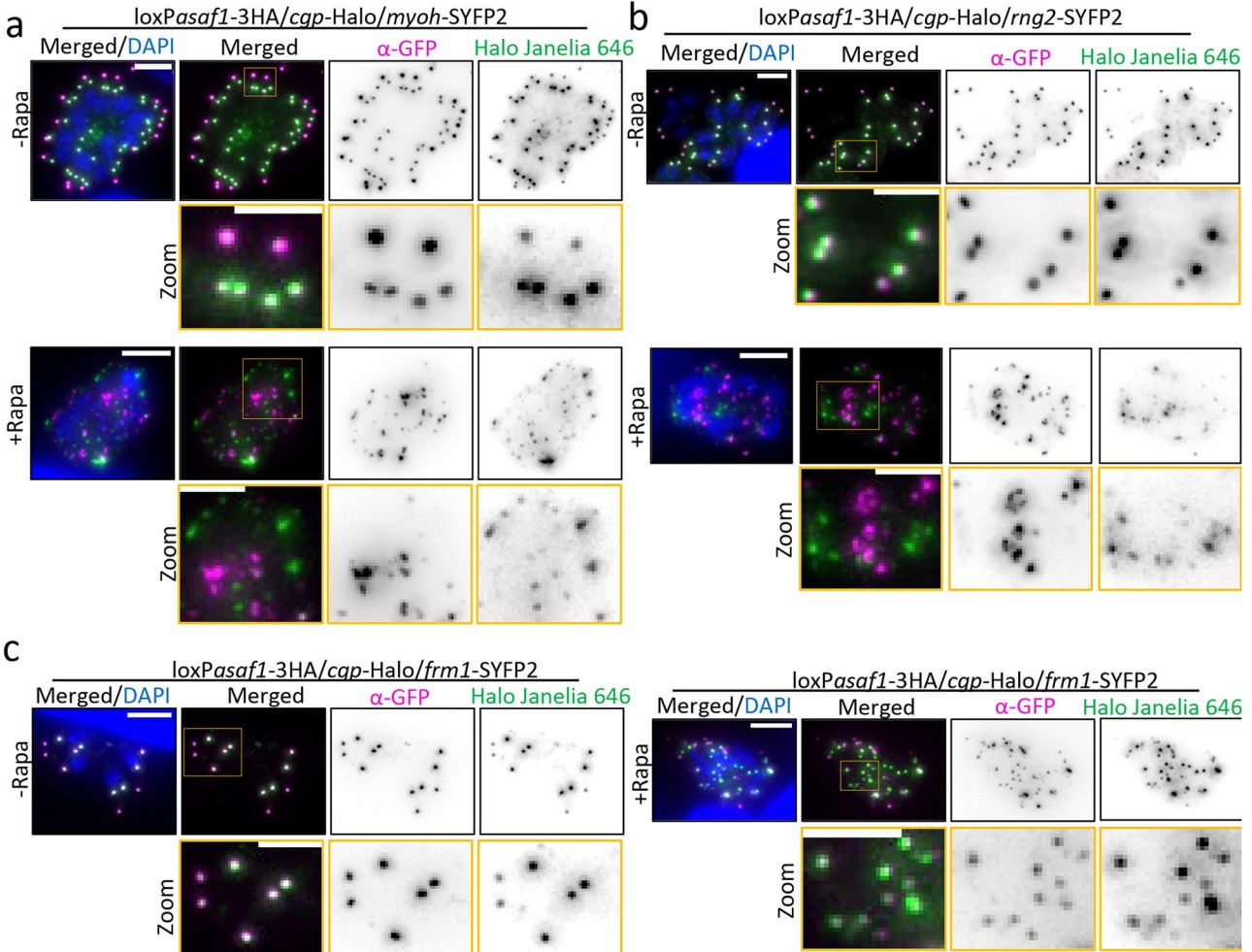

**Fig. 6 | ASAF1 depletion disrupts the organization of the conoid complex.**
**a** Effect of ASAF1 depletion on sub-structures of the nascent conoid (*n* = 2). Upon ASAF1 depletion, CGP (marker of PCRs) and MyoH (conoid) dissociate from each other and show no colocalization. Both markers appear randomly dispersed in the cytosol. Scale bar: 5 μm. b, Effect of ASAF1 depletion on CGP and RNG2 (APR marker) (*n* = 2). Upon ASAF1 depletion, CGP and RNG2 dissociate from each other, similar to (**b**). Scale bar: 5 μm. **c** Effect of ASAF1 depletion on CGP and FRM1 (*n* = 2). CGP and FRM1 remain associated with each other upon ASAF1 depletion, although both are randomly distributed in the cytosol. Scale bar: 5 μm.

matures and ASAF1 expression ceases leaving CGP, Pcr4, and 5 to stabilise the PCR.

## Methods

### T. gondii and host cell culture
*T. gondii* tachyzoites were cultured on confluent human foreskin fibroblasts (HFFs; ATCC, SCRC-1041) at 37 °C and 5% $CO_2$ using DMEM (Sigma, D6546) supplemented with 10% fetal bovine serum (BioSell, FBS.US.0500), 4 mM L-glutamine (Sigma, G7513), and 20 μg/ml gentamicin (Sigma, G1397).

### Generation of transgenic parasites
The CRISPR/Cas9 system was used to generate transgenic parasites. Guide RNAs targeting the gene of interest were designed using EuPaGDT[45] and are listed in Supplementary Table 2. The sgRNAs were ligated into a vector coding for Cas9-YFP expression as previously described[46].

Repair templates for integrating a tag and a loxP sequence were generated following the method described by Stortz et al.[24,47,48]. Briefly, for tagging, the repair templates were PCR-amplified from vectors carrying tags such as 3xHA, SYFP2, Halo, and SNAP. Primers used for this process were designed to bind the vectors and included 50 bp of homology to the gene. For the upstream loxP, oligonucleotides

containing the loxP sequence, flanked by 33 bp of homology on both sides of the gene, were ordered as single-stranded DNA from Thermo Fisher.

The HaloTag system was utilized to label CGP and other proteins due to its versatility in conjugating different fluorophores and its ability to provide high signal-to-noise ratios and specificity when combined with Janelia dyes (Promega). This approach was particularly advantageous for imaging low-expressing proteins.

Parasite transfection, sorting, and screening for positive clones were performed as described by Stortz et al.[47]. In short, repair templates and Cas9-YFP expression vectors were co-transfected into freshly lysed or mechanically lysed RHDiCre Δku80[49] tachyzoite parasites. After 24-48 hours post-transfection, parasites were sorted into 96-well plates using FACS (FACSARIA III, BD Biosciences). Single parasite clones were isolated and examined via genomic PCR to confirm correct modifications and sequenced where necessary. All insertion PCR gel images are available from the authors upon request.

When indicated in the figure legend, TLAP1 (TrxL1-associating proteins)[50] endogenously tagged was used to show parasites in replicating and non-replicating stages.

Primers are listed in Supplementary Table 3, and the transgenic parasite lines generated/used in this study are listed in Supplementary Table 4.

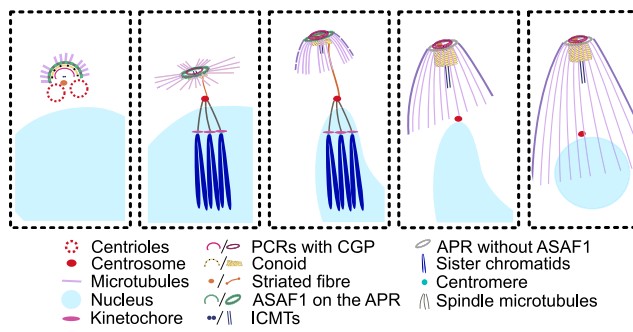

| | | |
|---|---|---|
| ◌ Centrioles | PCRs with CGP | APR without ASAF1 |
| ● Centrosome | Conoid | Sister chromatids |
| — Microtubules | •/▪ Striated fibre | • Centromere |
| Nucleus | ASAF1 on the APR | ║ Spindle microtubules |
| — Kinetochore | •/║ ICMTs | |

**Fig. 7 | Model of conoid complex formation.** ASAF1 is recruited early after centriole duplication to define the apical polar ring (APR) and organize subpellicular microtubule assembly. During daughter cell development, ASAF1 acts as a scaffold for ordered conoid complex formation. After maturation, CGP stabilizes the preconoidal rings (PCRs) and anchors FRM1 to enable motility initiation. Loss of ASAF1 blocks daughter cell formation, whereas CGP depletion destabilizes mature PCRs and disrupts gliding motility.

To induce KO, 50 nM rapamycin was added for one to two hours, followed by a wash with fresh DMEM media, and parasites were allowed to grow until fixation.

### Proximity labelling and biotinylated protein purification

Biotin labelling and subsequent processing were performed according to Singer et al.[51], with some modifications. Briefly, after 24 h post-infection, parasites (*cgp*-TurboID/*frm1*-3HA, *frm1*-TurboID/loxP*cgp*-Halo, and DiCreΔku80) were treated for 6 h with or without 150 μM biotin. They were then mechanically released, filtered through 3 μm filters, and washed three times with cold PBS. A total of $6 \times 10^7$ parasites were pelleted to purify biotinylated proteins. Additionally, $10^7$ parasites treated with ± 150 μM biotin were collected for biotinylation analysis via Western blot. All harvesting steps were performed on ice or at 0–4 °C. Parasite pellets were kept at −80 °C before being used in subsequent steps.

To purify biotinylated proteins, $6 \times 10^7$ parasites were lysed for 30 minutes in 950 μL of RIPA buffer (0.5% sodium deoxycholate, 150 mM NaCl, 1 mM EDTA, 0.1% SDS, 50 mM Tris-HCl pH8.0, 1% Triton TX-100) with Pierce™ protease inhibitor. The supernatant from the lysis was incubated with 50 μL of beads per sample (Dynabeads™ MyOne™ Streptavidin T1, Invitrogen), prewashed with PBS, for 30 min at room temperature while being gently rotated. The beads were then washed five times with 1 mL of RIPA buffer without Triton TX-100 but with protease inhibitor, followed by three washes with 50 mM Tris-HCl (pH 8). Washing steps were performed in a cold room. While 10% of the beads were preserved for Western blot analysis, the rest of the beads were pelleted and stored at −80 °C before being sent for mass spectrometry. In total, samples were harvested and processed independently three times (*n* = 3 biological replicates).

### Protein quantification by LC-MS

Immunoprecipitated proteins were quantified as described before with minor changes[51]. Briefly, the beads were treated with 10 ng/L of trypsin in 1 M urea and 50 mM $NH_4HCO_3$ for 30 min, then rinsed with 50 mM $NH_4HCO_3$. Afterwards, the supernatant was digested overnight with 1 mM DTT. Before LC-MS analysis, the peptides were alkylated and desalted after digestion.

For LC-MS/MS, the peptides were injected into an Ultimate 3000 RSLCnano system and separated in a 25 cm analytical column (75 μm ID, 1.6 μm C18, Aurora-IonOpticks) with a 50 min gradient from 2 to 35% acetonitrile in 0.1% formic acid. The effluent from the HPLC was directly electrosprayed into a Qexactive HF instrument operating in data-dependent mode to automatically transition between full-scan

mass spectrometry (MS) and MS/MS acquisition. Survey full-scan MS spectra (from m/z 375–1600) were acquired with resolution R = 60,000 at m/z 400 (AGC target of $3 \times 10^6$). The 10 most intense peptide ions with charge states between 2 and 5 were sequentially isolated to a target value of $1 \times 10^5$ and fragmented at collision energy normalised to 27%.

MaxQuant 2.0.1.0 was used to identify and quantify proteins using iBAQ with the following parameters: Uniprot_UP000005641_Toxoplasmagondii_20201123.fasta; MS tol: 10 ppm; MS/MS tol: 20 ppm Da; Peptide FDR: 0.1; Protein FDR: 0.01 min. peptide length: 7; Variable modifications: Oxidation (M); Fixed modifications: Carbamidomethyl (C); Peptides for protein quantitation: razor and unique; Min. peptides: 1; Min. ratio count: 2.

The quantified proteins (MaxQuant iBAQ Z-score normalised values) were compared using the adjusted *t*-test function from the Volcano plot option from Perseus 1.6.15.0 (missing values from the normal distribution replaced, width: 0.3 and downshift: 4, the false discovery rate (FDR): 0.05 and the S0 value: 0.1.

### Thin-section TEM of Epoxy-embedded samples

For loxP*asaf1*-SYFP2, the parasites were induced for 24 h with or without rapamycin, released mechanically prior transfer to Ibidi μ-dishes previously seeded with HFF cells. After 24 h of replication, the parasites were fixed with 2.5% glutaraldehyde in 0.1 M phosphate buffer pH 7.4. The parasites were washed three times at room temperature with PBS (137 mM NaCl, 2.7 mM KCl, 10 mM $Na_2HPO_4$, 1.8 mM $KH_2PO_4$, pH 7.4) and post-fixed with 1% (w/v) osmium tetroxide for 1 h. Subsequent to washing with PBS and water, the samples were stained en bloc with 1% (w/v) uranyl acetate in 20% (v/v) acetone for 30 min. Samples were dehydrated in a series of graded acetone and embedded in Epon 812 resin. Ultrathin sections (thickness 60 nm) were cut using a diamond knife on a Reichert Ultracut-E ultramicrotome. Sections were mounted on collodium-coated copper grids, post-stained with lead citrate (80 mM, pH 13) and examined with an EM 912 transmission electron microscope (Zeiss, Oberkochen, Germany) equipped with an integrated OMEGA energy filter operated in the zero-loss mode at 80 kV. Images were acquired using a 2k × 2k slow-scan CCD camera (Tröndle Restlichtverstärkersysteme, Moorenweis, Germany). Pixel is 0.0014 um.

### Sample preparation for in situ cryo-CLEM

Human foreskin fibroblasts infected with *T. gondii* parasites from the strain loxP*cgp*-Halo/pcr4-Halo were pretreated with +/− 50 nM rapamycin for 72 h. Cells were stained with 0.04 μM HaloTag Janelia 646 dye for 1 h at 37 °C, scratched with a cell scraper, and syringed to release the parasites. Extracellular parasites were resuspended in PBS and stained with NucBlue (ThermoFisher R37605) according to manufacturer's instructions for 15 min at 37 °C. The parasites were washed with PBS and incubated with +/− 2 μM calcium ionophore (Sigma-Aldrich A23187). The parasites were counted and resuspended in PBS to a concentration of around $8 \times 10^7$/mL. Quantifoil lacey EM grids were plasma cleaned for 30 s with 90:10 argon-oxygen gas mixture with a Fischione 1070 plasma cleaner set to 100% power and flow of 30 SCCM right before plunge freezing. 4 μL of parasites were dispensed on the EM grid and plunge-frozen into liquid ethane with a Leica GP1 plunger after 3–4 s of backside blotting. The plunger was set to 22 °C and 99% humidity. The vitrified grids were clipped into AutoGrid cartridges and stored in liquid nitrogen.

### Cryo-light microscopy

Brightfield imaging under cryogenic conditions was performed with a Zeiss LSM900 upright microscope equipped with a Linkam cryo-stage. The grids were loaded on the cryo-stage, and an atlas of the full grid was taken with a 5x dry objective with NA = 0.2 at brightfield, 353 nm and 653 nm. The acquired grid atlas was used to assess the ice

thickness and the parasite distribution throughout the grid. Subsequently, a 800 × 1000 μm tile scan of the middle of the grid was taken using a 100x NA = 0.75 objective with Z-stack spanning across 5–7 μm in the same channels. Maximum-intensity projection and stitching of the datasets were performed using ZEN 3.8 software (Zeiss).

## Cryo-electron tomography

The clipped cryo-grids were loaded onto Krios TEM operated at 300 kV equipped with a Falcon 4i direct electron detector (ThermoScientific) and Selectris X energy filter. Tilt series were collected by means of SerialEM v4.1.0[52] and acquired at a nominal magnification of 42,000x corresponding to a pixel size of 3.03 Å. A 50 μm C2 and a 100 μm objective aperture were inserted and the width of the energy filter slit was set to 10 eV. The TEM was operated in nanoprobe mode at spot size 5. Images were acquired using a dose-symmetric tilt scheme with 2° step and a total range of −60° to 60°. Each movie was acquired with an exposure time of 1.2 s and a fluence (electron dose) of ~2.3 e-/Å$^2$ per tiltresulting in a total fluence (total dose) of ~140 e-/Å$^2$ for the full tilt series.

## Tomogram reconstruction

Acquired tilt series were reconstructed using command line version of Warp pipeline (v2.0.0dev29)[53] and denoised using IsoNet (v0.2.1)[54].

## Segmentation of tomograms

Reconstructed tomograms were segmented manually using Microscopy Image Browser (MIB, v2.84)[55] with the focus on the conoid complex and visualised using ChimeraX (v1.8)[56].

## Correlation of cryo-light and electron microscopy data

Cryo-fluorescence microscopy images were correlated with transmission electron microscopy (TEM) montages acquired at a pixel size of 5.6 nm using the ec-CLEM(v2) plugin[57] in Icy software (v2.5.4.0)[58]. Correlation was achieved by identifying easily recognizable features of the lacy EM grid support – such as the edges of irregular holes – visible in both the light and electron microscopy images. These features served as 2D fiducial markers. Approximately 15–20 corresponding points were identified in both the stitched fluorescence overview and the TEM montage, enabling precise image correlation within the software.

## Immunofluorescence assay

Parasites were fixed with 4% paraformaldehyde (PFA) or methanol at room temperature for 15–20 min. Samples were then blocked and permeabilized with 2% bovine serum albumin (BSA) containing 0.2% Triton X-100 in PBS solution for at least 20 min. Primary and 5secondary antibodies (Supplementary Table 5) were used to label proteins for 1 h and 45 minutes, respectively. After each labeling, samples were washed three times with PBS. Coverslips were mounted with ProLong Gold Antifade Mountant (Thermo Fisher Scientific P36930) or ProLong™ Gold Mountant with DAPI (Thermo Fisher Scientific, P36931).

For the detection of SYFP2-tagged proteins, the choice between direct visualization via SYFP2 fluorescence and the use of anti-GFP antibodies was based on the natural expression levels of each protein. Proteins with sufficiently high expression levels were visualized directly, while lower expression proteins were detected with antibodies to enhance the signal-to-noise ratio for improved imaging clarity. This was indicated in the figure legends as needed.

## Parasite labelling with dyes

To visualize Halo-tagged proteins, parasites were incubated with either 20 nM HaloTag Janelia 646 (Promega, GA112A), 20 nM HaloTag Janelia 549 (Promega, GA111A), or 200 nM HaloTag Oregon Green (Promega, G280B) for 1–2 h prior to fixation. For colocalization analysis, 50 nM HaloTag Janelia 549/646 was used. Sequential labelling of both Halo-

and SNAP-tagged parasites, including loxP*cgp*-Halo/*pcr4*-SNAP, loxP*pckmt*-Halo/*pcr4*-SNAP, and loxP*cgp*-Halo/*frm1*-SNAP/*imc1*-SYFP2, began with an initial incubation of 250 nM SNAP-Cell® 647-SiR (SiR-SNAP) (Biolabs, S9102S) for 1 hour, followed by labelling with 50 nM HaloTag Janelia 549. After each dye application, parasites were washed three times with PBS and incubated in media for 10 min prior to fixation or a second labelling. For loxP*cgp*-Halo/*frm1*-SNAP parasites, 100 nM HaloTag Oregon Green was used to label CGP, and 1 μM SiR-SNAP was simultaneously used to stain FRM1 for one hour, followed by washing and a one-hour media incubation before fixation.

## Plaque assay

A total of 500 parasites per well were inoculated into confluent HFFs in 6-well plates and cultured for 6 or 7 days undisturbed, either with or without 50 nM rapamycin. The HFF monolayer was washed once with PBS prior to fixation with ice-cold methanol for 20 min. The HFFs were stained using Hemacolor Rapid Staining of Blood Smear Solution 2 for 30 seconds, followed by Solution 3 for 2 min, and then washed three times with PBS. Images were captured using LAS X Navigator software on a Leica DMi8 Widefield microscope with a 10× objective, as previously described in ref. 24. Approximately 30 plaques from each of the three biological replicates were measured using ImageJ. Plaques were outlined manually using the drawing tool, and the area of each plaque was quantified.

## Rhoptry secretion assay based on ROP16-mediated STAT6 phosphorylation

ROP16-mediated STAT6 phosphorylation was assessed as previously described[59], with minor modifications. loxP*cgp*-Halo and DiCreΔKu80 parasites were incubated with or without rapamycin for 72 h prior to the rhoptry secretion assay. Parasites were mechanically released, pelleted, and resuspended in complete *T. gondii* culture medium. A total of 5 × 10$^5$ parasites were added to each well of 24-well plates containing coverslips with confluent HFFs. To facilitate settling onto the host cell monolayer, plates were incubated on ice for 10 min and subsequently centrifuged at 250 × g for 1 min. Plates were then incubated at 37 °C for 30 min before fixation with ice-cold methanol for 8 min. IFA were performed using anti-phospho-STAT6 and anti-SAG1 antibodies[60]. DAPI was used to stain the nuclei. The total number of nuclei and phospho-STAT6-positive nuclei were quantified by counting in approximately 20 randomly selected fields per coverslip. Three biological replicates were performed, each with two technical replicates.

## Expansion microscopy (ExM)

ExM was performed as described by Dos Santos et al.[61] with some modifications. Briefly, intracellular parasites on 12 mm coverslips were fixed with 4% PFA, followed by three washes with PBS, and stored at 4 °C. The fixed coverslips were then treated with a 1.4% formaldehyde and 2% acrylamide mix at 37 °C for 5 h, followed by a gelation step. A 35 μL drop of Monomer Solution (19% sodium acrylate, 10% acrylamide, 0.1% bis-acrylamide) supplemented with 0.5% Tetramethylethylenediamine (TEMED) and 0.5% Ammonium persulfate (APS) per coverslip was used for gelation on ice in a humid chamber for 5 minutes, then at 37 °C for 1 h. After gelation, the gel was denatured with denaturation buffer (200 mM SDS, 200 mM NaCl, 50 mM Tris in ultrapure water, pH 9) at 80/85 °C for 1.5 hours. Gels were then expanded overnight in deionized water (dH$_2$O) at 4 °C. A small piece of the gel was cut and subjected to immunostaining. For immunostaining, antibodies were diluted in freshly prepared PBS with 2% BSA. The gel was incubated with primary antibodies for 3 hours at 37 °C, followed by four washes with PBS containing 0.2% Triton X-100 (PBS-Tx100). The gel was then incubated with secondary antibodies for 2.5 h at 37 °C or overnight at 4 °C, followed by an additional 2 h at 37 °C, and then washed four times with PBS-Tx100. Antibody concentrations are listed

in Supplementary Table 5. The gel was fully expanded in dH$_2$O for a few hours or overnight. To stain the nuclei, 0.8 μM Hoechst 33342 (diluted in dH$_2$O) was then used for 3 min, followed by one wash. After determining the orientation of the gel, it was mounted on a glass bottom dish pre-coated with poly-L-lysine. Widefield images were acquired in z-stacks with 0.3 μm increments using a Leica DMi8 wide-field microscope.

## Protrusion assay

LoxP*cgp*-Halo/*frm1*−3HA and DiCreΔKu80 parasites were pretreated with or without rapamycin for 72 h prior to the assay. Parasites were mechanically released, filtered, pelleted, and resuspended in 1 mL pre-warmed PBS containing 4 μM calcium ionophore A23187 (Sigma-Aldrich, C7522-1mg). After incubation at 37 °C for 10–12 min to promote extrusion, parasites were fixed with PFA. The fixed parasites were transferred to 24-well plates containing poly-L-lysine-coated coverslips and subjected to brief centrifugation at -161 × g for 1 min. The parasites were then processed for ExM and stained with anti-acetylated tubulin, which was used for both strains, and anti-HA antibodies, specifically for the loxP*cgp*-Halo/*frm1*−3HA strain. Three biological replicates were performed, with at least 100 parasites counted per condition and replicate. For conditional knockouts of CGP, only vacuoles lacking HA signal in the apical region were counted, as excision of CGP results in the absence of FRM1 at the PCR. Additionally, the Halo antibody did not work effectively for the CGP-Halo strain.

## Western blot

Parasite strains were treated with or without rapamycin for 48 h (loxP*asaf1*-3HA/*myoh*-SYFP2) or 72 h (loxP*cgp*-Halo/*pcr4*-SYFP2 and loxP*cgp*-Halo/*icap16*-SYFP2) before harvest and subsequent Western blot analysis.

For Western blotting, 4–15% precast polyacrylamide gels (Bio-Rad, 4561083) were used. Membranes were probed with antibodies against aldolase, GFP, or HA, as summarized in Supplementary Table 5. The Chameleon Duo Pre-stained Protein Ladder (LI-COR, 928-60000) was used as the molecular weight marker. Stained membranes were imaged using the Odyssey CLX-1849 system (LI-COR).

For quantification, all experiments were performed in at least three biological replicates, with samples run on independent membranes. Band intensities were measured using ImageJ. Quantification graphs represent the normalized percentage of MyoH, Pcr4, or ICAP16 levels, first normalized to aldolase as a loading control and then calculated relative to the minus rapamycin condition.

## Image acquisition and image analysis

Widefield images were taken using a Leica DMi8 wide-field microscope. Widefield images, except for ExM images and images in Fig. 5a, b, Supplementary Fig. 3b, 5a (210430 candidate) and 10a were deconvolved using Huygens Essential v.18.04. ExM images were contrast-adjusted for better visualization.

For confocal and STED imaging, images were acquired on an Abberior 3D STED microscope using confocal mode or 2D STED mode. Images were processed with Fiji (ImageJ) software v.2.3.0.

## Time lapse microscopy

All live microscopy was performed using a Leica DMi8 widefield microscope equipped with a DF C9000 GTC camera in a heated chamber with 5% CO$_2$.

To investigate CGP behaviour during replication, freshly egressed CGP-Halo/IMC1-SYFP2 parasites were pelleted and resuspended in media containing 20 nM Janelia 646 dye. The parasites were allowed to invade confluent HFFs in a 35 mm imaging dish with a polymer coverslip bottom for 20 min, followed by washing with warm PBS to remove non-invaded parasites. FluoroBrite DMEM media, supplemented with 10% foetal bovine serum, 4 mM L-glutamine, 20 μg/ml

gentamicin, and 4 nM Janelia 646, was used during video recording. Time-lapsed microscopy started approximately 30 min post-inoculation and continued overnight using a 100x oil objective lens. Images were acquired at 30 min intervals, with a z-stack of 6 μm in 1 μm increments.

To examine CGP behavior during gliding, time-lapsed microscopy was performed according to the method described by Li et al.[24], with slight modifications. Freshly egressed CGP-Halo/IMC1-SYFP2 parasites were labelled with 10 nM Halo Janelia 646 for at least two hours. Parasites were then resuspended in endo buffer (44.7 mM K2SO4, 10 mM MgSO4, 100 mM sucrose, 5 mM glucose, 20 mM Tris, 0.35% w/v BSA, pH 8.2) and allowed to settle on an FBS-coated glass-bottom dish for 10 minutes. The media was then changed to warm gliding buffer (1 mM EGTA and 100 mM HEPES in HBSS solution) prior to imaging. Images were captured at 0.5 frames per second (FPS) using brightfield and far-red channels with a 100x oil objective lens with z-stacks.

To investigate CGP behaviour during egress, CGP-Halo/IMC1-SYFP2 parasites were grown in HFFs on a glass-bottom dish for approximately 30 h before induction of egress with 2 μM Calcium ionophore A23187. Images were taken at 0.5 FPS using the green and far-red channels in auto-focus mode with a 100x oil objective lens.

To examine the effect of ASAF1 deletion on IMC1 or tubulin during replication, similar procedures were followed as described above. However, loxP*asaf1*-SYFP2/IMC1-mCherry parasites were used to infect HFFs for 30 min, and video recording started around 8 h post-inoculation, focusing on the red fluorescence channel with adaptive autofocus mode. In the case of rapamycin addition, it was added at the beginning of the inoculation period. For imaging the effect on tubulin, vectors containing tubulin-mCherry under the tubulin promoter were transfected into loxP*asaf1*-SYFP2 parasites. After overnight growth, rapamycin was added to induce KO, and imaging started around 8 h post-induction. Non-rapamycin-treated conditions were also recorded at a similar time point. Images were taken at 30-minute intervals, focusing on the IMC1/tubulin channel in auto-focus mode with a 100x oil objective lens.

All videos were recorded in triplicate per condition at a minimum.

## Software and data analysis

In silico cloning was performed using ApE - A plasmid editor (by M. Wayne Davis, v.2.0.53c) software. Statistical analysis and graphs were generated by Graphpad Prism 8.2.1. Sequencing results were analysed using BioEdit v.7.2. or ApE Generation of gRNAs for tagging was done in EuPaGDT31. Schemes were created using Inkscape v.1.3.2 or Microsoft PowerPoint. We used ChatGPT (OpenAI) to assist with improving clarity and grammar during manuscript preparation. All scientific content, data interpretation, and conclusions were generated by the authors.

## Reporting summary

Further information on research design is available in the Nature Portfolio Reporting Summary linked to this article.

## Data availability

The mass spectrometry proteomics data generated in this study have been deposited to the ProteomeXchange Consortium via the partner repository[62] with the dataset identifier PXD053452. Source Data are provided within this paper. Source data are provided with this paper.

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

## Acknowledgements

We thank all colleagues, who contributed antibodies and reagents for this study. We thank the Mattei lab members for discussion and technical support. We thank Mirko Singer for generating the vectors for tagging and inserting the 5′ loxP site of FRM1, as well as for generating the loxP*frm1*-mCherry strains. W.L. is funded via a CSC fellowship (201806910075; W.L.) and Bavarian Gender Equality Grant (W.L.). This project is funded within the DFG Priority Programme SPP2225 EXIT strategies of intracellular pathogens, (ME 2675/7-1 and JI 463/2-2; M.M and E.J-R.) and a DFG Equipment grant INST 86/2308-1 (M.M.).

## Author contributions

W.L. characterized CGP, performed BioID, identified and conducted phenotypic assays for ASAF1, analyzed the data, and wrote the paper. O.K. performed cryo-ET, analyzed the cryo-ET data and wrote the cryo-ET results section. P.Q. assisted with the localization of candidates from the TurboID candidate list. I.F. performed the LC-MS and conducted the relevant analysis. S.G. performed the TEM imaging and analysis. J.G. prepared the samples for TEM. A.K. provided access to the EM facility and contributed resources. S.M. analyzed the cryo-ET data and contributed resources. E.J-R. designed and coordinated the project and experiments, analyzed the data, and wrote the paper. M.M. designed and coordinated the project and experiments, analyzed the data, contributed resources, and wrote the paper.

## Funding

## Competing interests

The authors declare no competing interests.
