## [Transparent Peer Review file · Nature Communications]

An apical ring protein essential for conoid complex assembly and daughter cell formation in *Toxoplasma gondii*

Corresponding Author: Professor Markus Meissner

Version 0:

Reviewer comments:

Reviewer #1

(Remarks to the Author)

Background:

CGP is a crucial protein of the preconoidal rings, initially identified in a splitCas9 phenotypic screen by the Meissner/Jimenez-Ruiz group (Li, Grech, Stortz et al., Nat Micro. 2022). At that time, CGP was shown to play a role in actin-based motility, significantly affecting invasion and egress. Around the same period, the Soldati-Favre group characterised various preconoidal rings proteins, including the essential motility factor Formin1 (FRM1), which was observed to localise to the preconoidal rings (Dos Santos et al., Nat Micro. 2022). Notably, certain mutants of preconoidal rings proteins, specifically Pcr4 and Pcr5, have been found to be essential for maintaining the stability of the preconoidal rings.

In a separate development, recent reports from various teams have provided a detailed account of the early stages of daughter cell formation, using advanced high-resolution microscopy techniques (Arias Padilla et al., MBoC, 2023 ; Arias-Padilla, Munera Lopez et al. JCS. 2024 ; Engelberg et al., Biorxiv, 2024 ; Haase et al., MBoC, 2024). Some of those reports finally reported on the initial role of gamma-tubulin in the early development of daughter cells scaffolds. While gamma-tubulin is crucial for microtubule polymerisation in other organisms, it has never been observed at the apical polar ring, which is considered the microtubule organising centre (MTOC) in apicomplexan parasites.

Findings:

In this study by Li, Koczy et al., the role of CGP is examined in detail. The authors demonstrate that CGP is essential for maintaining the stability of the preconoidal rings in mature parasites, though not in daughter cells. The absence of CGP mirrors the effects of Pcr4/5 depletion, resulting in the specific loss of preconoidal ring proteins. Consequently, the severe motility, invasion, and egress phenotypes observed in previous studies can now be attributed to the disruption of the preconoidal rings and loss of FRM1.

In the second part of the manuscript, the authors conduct a proximity labelling experiment to identify new conoid complex proteins, using CGP and FRM1 as markers. While previously known proteins of the complex are identified in both CGP and FRM1 BioID assays, other, as yet undescribed proteins are also detected. The authors particularly highlight ASAF1, an essential protein located at the apical polar ring in developing daughter cells. Depletion of ASAF1 results in a marked defect in daughter cell formation, though it does not impact the formation of key structures such as centrosomes, kinetochores, and the striated fibre. Finally, the authors demonstrate that ASAF1 depletion disrupts the assembly of the conoid complex and subpellicular microtubule scaffold, thereby abrogating daughter cell formation.

General comments:

This study is of high quality, presenting an impressive volume of data and a broad range of methods. However, while the depth of data is remarkable, the manuscript's narrative could be more cohesive. It currently features two main threads: firstly, the continuation of the CGP characterisation begun in the authors' previous study (Li, Grech, Stortz et al., Nat Micro. 2022), and secondly, the characterisation of ASAF1, a protein involved in daughter cell formation.

The first part, covering the characterisation of CGP, feels somewhat lengthy and complex, spanning three main figures when it might be more effectively conveyed in one or two. Presenting the cryo-ET data illustrating the loss of the preconoidal rings

in the absence of CGP as the initial figure could make the current Figure 1 redundant, possibly placing it better as supplementary material. Moreover, since the authors already established in their prior publication that CGP is indeed a resident protein of the preconoidal rings, proving this point again here is unnecessary.

I find the second part of the manuscript, focusing on ASAF1 and its role in daughter cell formation, particularly engaging. These findings are also well-timed, aligning with recent publications. The super-resolution microscopy, expansion microscopy, and video microscopy provide compelling insights into daughter cell formation. However the author should be more careful about the use of the terms 'conoid' instead of 'conoid complex' or 'APR' which sometimes makes some conclusions difficult to understand.

Specific comments (in order of appearance within the text):

- In the text, it would be preferable if the author could avoid the term "apical rings," as this might cause confusion for readers who are not specialists in the field. The terms "apical polar rings" and "preconoidal rings" refer to specific, well-defined structures and should ideally be used as such. Along the same lines, the term "conoidal" may benefit from more precise usage. It is certainly appropriate to describe a protein as "conoidal" if it localises to the conoid. However, it may be more accurate to refer to a protein within the broader apical complex as an "apical complex protein," rather than "conoidal." This is especially relevant to the naming of CGP (Conoid Gliding Protein), which, in my view, could benefit from reconsideration, as it is not a conoid protein but a preconoidal rings protein and is involved in functions beyond gliding, which may lead to some ambiguity.
- In the introduction, particularly in the first and second paragraphs, it would be helpful for the authors to more clearly distinguish between aspects specific to Apicomplexa in general and those specific to Toxoplasma. For instance, while it is accurate that the conoid complex contains 'unique secretory organelles' across all Apicomplexa, it should be noted that Plasmodium and Cryptosporidium do not possess ICMTs within the conoid. Rephrasing these sections to clarify what constitutes general information about Apicomplexa and what is specific to Toxoplasma would enhance the reader's understanding of the manuscript's particular focus.
- In the introduction, it may be helpful to specifically outline the proposed model for the formation of the SPMTs basket as described by Engelberg and colleagues. While this model is explained later in the text, providing a clear summary in the introduction could improve reader comprehension. I am specifically referring to the initial formation of microtubules near the centrioles, which are then subsequently incorporated into the developing APR.
- Page 3 line 19-20: rephrase "with the unconventional myosin MyoH localized at the conoid" instead of "apical localised unconventional myosin MyoH" for clarity.
- page 5 line 1: all abbreviations are already defined in the introduction and can be used without writing the name of the proteins GAC, AKMT and MyoH in full.
- page 5 line 4: It would be helpful if the author could provide a bit more explanation about the Halo tag, including what it is and why it has been utilised in this study.
- Figure 1B: The observation that MyoH is unaffected in the absence of CGP appears somewhat redundant, given that the previous study already demonstrated that neither apical polar ring (APR) nor conoid proteins were impacted by CGP depletion.
- Figure 1B-C-D: Could the author clarify why certain SYFP2-tagged proteins are detected directly via SYFP2 fluorescence (MyoH / GAC), while others require detection with an anti-GFP antibody (AKMT)?
- Figure 2A: I am not entirely convinced by the STED images throughout the manuscript. For instance, in Figure 2A, it's unclear how much additional resolution STED provides over the confocal imaging. Given the authors' evident proficiency with expansion microscopy and the significantly higher gain in resolution it offers, I believe it would be beneficial to use expansion microscopy in place of STED. Additionally, as previously mentioned, demonstrating that CGP is a preconoidal ring protein feels somewhat redundant, as this was already convincingly shown in the previous study, where the co-localisation with SAS6L/RNG2 was particularly clear.
- Page 5 line 24: Could the authors please clarify what is meant by the term 'second apical polar ring'? Are there multiple apical polar rings present and therefore several MTOCs? What are the several APRs described throughout the text?
- Figure 2: Given the author's expertise in using fluorescent markers and video microscopy, it would be advantageous to conduct video microscopy on preconoidal ring-tagged lines when CGP is downregulated. This would provide critical data regarding the timing of preconoidal ring loss during daughter cell hatching. Capturing video footage with images taken every 5 to 10 minutes and spanning a full cell cycle would yield highly compelling results. It is unfortunate that not many researchers in the field are utilising video microscopy, and I salute the author for their efforts in this area.
- Page 6 line 19-21: The sentence "We speculate that this stabilisation step requires the contact of PCR proteins with the plasma membrane which only occurs after hatching of the daughter cells from the mother" would be more appropriately placed in the discussion section, as there is currently no supporting evidence.

- Page 7 line 21-23: Could the authors explain why some parasites are still able to carry out conoid protrusion even in the absence of pre-conoidal rings? This observation appears to challenge the current model regarding conoid protrusion and the role of actin in this process. Additionally, what is meant by the phrase 'in some cases'? Providing quantification for these observations would be extremely beneficial.
- Figure 3: I am unable to locate the number of parasites reconstructed for the cryo-electron tomography under each condition. It would be beneficial for the authors to include quantification to support their conclusions regarding the integrity of the ICMTs, APR, and MVs.
- Page 7 (line 28-29) and page 8 (line 2-3): Was the disruption of the rhoptries anticipated in the absence of pre-conoidal rings? This appears to be a rather surprising finding. Is the apical vesicle (AV) (Mageswaran et al. Nat Comm. 2021; Segev-Zarko et al. PNAS Nexus. 2022) still present in absence of the PCRs? If this observation holds true, it would be beneficial for the authors to strengthen their data by employing additional techniques that are more suited for quantification. Furthermore, conducting a rhoptry secretion assay would be valuable to demonstrate how the loss of pre-conoidal rings impacts invasion, in addition to its effects on motility.
- Page 8 line 18-19: I find the phrase 'known or predicted localization at the apical region or unknown localizations predicted by HyperLOPIT' a bit unclear. Could the authors clarify the criteria they used for this classification? As it currently reads, it seems that 'predicted apical localisation' or 'unknown localisation' could refer to almost any location within the cell.
- Page 8 line 21: Please state clearly which proteins couldn't be tagged. Alternatively, this information can be added to the supplementary table 1.
- Extended data Fig.5F-G-H: Do these data suggest that ICAP16 is a resident protein of the PCRs? It might be more straightforward to confirm this through expansion microscopy.
- Extended data Fig.6A-B-C-D: Do these data suggest that DAP1 is a resident protein of the PCRs? It might be more straightforward to confirm this through expansion microscopy.
- Extended data Fig.7A-B-C-D: I suggest doing expansion microscopy instead of STED to prove the PCRs localisation of PCKMT.
- Page 9 line 18-20: A straightforward approach, such as expansion microscopy, to localise PCKMT, DAP16, and ICAP16 would provide clear confirmation of their positioning at the PCRs. This would also allow for more certainty in describing these proteins as being located at the PCRs, rather than stating that they 'appear to be localised' there.
- Page 9 line 19: Maybe replace "from the conoid" by "from the conoid complex/apical complex/apical pole of the parasite" to avoid confusion.
- Extended data Fig.8B: It would be helpful to clarify that TGGT1_238170 refers to IAP3, as this is not indicated in the text, table, or in the graph in panel A. This omission, in addition to the high amount of data and proteins investigated, makes it a bit challenging to follow the sequence of panels describing the numerous BioID candidates.
- Extended data Fig.8C: It would be helpful if the authors could provide an explanation of what TLP1/TLAP1 refers to, at least in the figure legend, as it is currently not described in the main text, figure legends, or supplementary figures. Additionally, primers used for tagging the gene do not appear to be specified. The dark blue signal shown by IFA is somewhat unclear in the images presented. If this refers to the TLAP1 protein identified by Ke Hu's team (Liu et al., Eukaryotic Cell, 2013), it would be useful to state this clearly and to reference their study.
- Extended data Fig.8F: Here, it may be more straightforward and precise to use expansion microscopy to co-localise ASAF1 with CGP, rather than relying on the current images. Additionally, it could be worthwhile to explore co-localisation with Pcr7 (Dos Santos et al., Nat Micro. 2022), which, to my knowledge, is the only protein described specifically at the pre-conoidal rings of daughter cells.
- Page 10 line 19: RNG2 is not a marker of the conoid but a marker of the APR.
- page 11 line 7-11: I have some reservations regarding the conclusions drawn here, particularly the statement that 'formation of the conoid and consequently for recruitment of additional APR proteins'. I am not entirely convinced that conoid formation is a prerequisite for the assembly of the APR. It seems plausible that the APR may actually form first, followed by the SPMTs and then the conoid. Additional experiments would be beneficial to support such a conclusion, as this would be particularly important for our understanding of the sequence in which the sub-compartments of the conoid complex are assembled. Furthermore, the claim that ASAF1 is essential for gamma-tubulin complex recruitment is not supported by the data presented here. It would be helpful if the authors could assess gamma-tubulin levels and localisation in the ASAF1-null mutant, and vice versa, to substantiate this point.
- page 11 line 26-27: I am not sure whether microtubules extending from the centrosomes, without any attachment to structures resembling an APR, should be referred to as 'subpellicular' microtubules.
- page 11 line 28-29: The APR and conoid are two different structures. The author should remove the conoid mention from

the sentence.

- Figure 5C: Do the author have any hypothesis/explanations regarding the 'membrane folds' observed in the ASAF1-null mutant?
- page 12 line 25: "ASAF1 is critical for formation of the conoid complex" or "ASAF1 is critical for formation of the APR" would be more precise.
- Figure 6A: Do the authors have any insight into the specific location within the conoid complex where SFA2 is anchored?
- Extended data Fig.10H-I: I'm a bit puzzled as to why the two graphs use different metrics. One graph examines dividing versus non-dividing, while the other focuses on the number of puncta for each protein. Could you help clarify the reasoning behind this difference? I think they should both be assessing the number of abnormal cytoplasmic puncta for each protein. It might have been interesting to also add Centrin1 to the list of protein of interest in this experiment as it has been shown to be fragmenting in gamma-tubulin depleted parasites.
- Figure 6B-D: Perhaps the authors might consider conducting a western blot analysis to assess the stability of the various proteins in the ASAF1-null mutant. This could help clarify the statement, "in the case they are still expressed" (page 13 line 14-15).
- Figure 6E: In the final schematic, the author should define what does 'apical rings' is referring to. If I'm not mistaken it should refer to the apical polar ring.
- page 13 line 29: 'huge' is quite subjective.
- Page 14 line 17: Could the author specify the data that supports the claim that 'contact of mature PCRs with the plasma membrane is required for its/their stability'? Despite being an appealing hypothesis, this claim seems to not be supported by any data presented here.
- Page 14 line 21: The rhoptry positioning data must be reinforced by additional experiments before drawing such conclusions.
- Page 15 line 4: Could the author perhaps clarify why they believe Pcr5 plays a lesser role in PCR stability compared to CGP or Pcr4?
- Page 15 line 11: In absence of CGP, no PCRs can be seen on mature cells. I think there is no 'mature PCR' in absence of CGP.
- Page 15 line 21: '...for the formation of daughter conoid complexes'

Reviewer #2

(Remarks to the Author)

The manuscript from Li W, Koczy O et al investigated in the apicomplexan parasite *Toxoplasma gondii* (*T. gondii*) the role of CGP, a protein localized at the pre-conoidal rings (PCR) and previously identified by the lab as essential for parasite motility. Inducible depletion of CGP combined with ultrastructural analysis (using cryo-electron tomography) showed that CGP is essential for stabilisation of the PCRs and recruitment of the F-actin nucleation protein FRM1 in mature conoid after daughter cell replication.

By using proximity-ligand binding assay with CGP as bait, they identified novel putative conoidal proteins possibly involved in conoid complex assembly. The authors further characterized in detail the function of ASAF1 and found that it is required for the early stage of conoid assembly in nascent daughter cells.

Depletion of ASAF1 resulted in failure of conoid complex assembly, disorganised subpellicular microtubules, and lack of IMC formation resulting in impaired daughter cell budding.

The conclusions drawn by the authors are supported by robust experimental work and convincing data sets. The results obtained are novel and interesting. In particular, the strength of the study lies on the combined use of high resolution (or ultrastructural) microscopy and live imaging. It is this spatio-temporal level that allowed the authors to identify new mechanisms of conoid complex assembly in *T. gondii*.

I have only minor comments.

Figure 1 :

Please provide Western blots confirming depletion of CGP upon rapamycin treatment and the consequence on the examined conoidal proteins by IFA. Notably, is total amount of FRM1 reduced in CGP-depleted parasites compared to control parasites ? Or depletion of CGP only impacts on FRM1 localization at the mature conoid ?

Line 9 : « Consequently, the defect in gliding motility upon CGP depletion results from the absence of FRM1-dependent F-actin nucleation at the apical tip in good agreement with previous studies ».

Please modulate the conclusion as the authors did not investigate and demonstrate that F-actin nucleation is impaired in CGP-depleted parasites.

Figure 2

STED-acquired signal for CGP does not display the expected resolution from STED, probably due to weak expression of this protein. Could the authors address the co-localization between PCR4 and CGP using expansion microscopy as they have done for FRM1 to confirm the localization of CGP to the PCRs.

Could the authors comment on the observation that upon CGP depletion, FRM1 is still recruited to conoids assembling in daughter cells in contrast to the conoid of the mother cell? In both structures, CGP is absent. This suggests that CGP is not required for FRM1 recruitment but only for mature PCR stability. Please comment this point in the discussion.

Fig 2E : as designed, the graph is difficult to read. Please modify it by indicating the percentage of CGP-positive conoids in mother versus nascent daughter parasites in both conditions : without and with rapamycin treatment. The authors may for example choose to examine only parasites in M phase (that include both mature and assembling conoids) to perform this quantification.

Line 19: « We speculate that this stabilisation step requires the contact of PCR proteins with the plasma membrane which only occurs after hatching of the daughter cells from the mother. »

Indeed, this hypothesis could likely explain why the PCRs are no longer present in the mature apical conoid of the mother cell.

Did the authors identify in their TurboID assay CGP partners that could contribute to PCR anchoring at the plasma membrane?

Figure 3

« However, in 33% of *cgp* iKO parasites, rhoptries could not be observed within the tomogram field of view (Fig. 3b and Extended Data Fig. 2; Supplementary Video 4) »

Please clarify : the absence of rhoptries within the conoid is a consequence of impaired anchoring at the apical tip of the parasite or is there a general defect in rhoptry formation ?

Consequently, please provide immunofluorescence images (expansion microscopy would be best) showing rhoptry formation and localization at different steps of parasite cell cycle in CGP-depleted parasites and provide quantification if any defect is observed.

Extended Data Fig. 3 / Table 1

Biotin labelling shows aspecific staining in what seems to be rhoptries or Golgi ? can the author clarify this point and comment.

Please provide a Venn diagram showing the number and identity of common versus unique proteins identified as partners of CGP versus FRM1 by TurboID.

The authors successfully tagged 13 candidate partners of CGP. Are these candidates common to FRM1 and CGP or unique to CGP?

Extended Data Fig 4

Many of the selected candidates are not localized at the apical tip of parasites or nascent conoids except for 293480 and 212780 (the latest being only present in nascent conoids). Could the authors comment on this important point and include a comment in the manuscript?

Extended Data Fig. 6h

Please quantify the plaque assay (also for extended Fig 9e)

As previously mentioned for FRM1, please investigate by WB whether absence of Dap1 and PCKMT signals at the mature conoid in CGP-depleted parasites is linked to their (at least partial) degradation ? or only mis-localization as written by the authors. If these proteins are degraded (knowing that dividing parasites with CGP signal in daughter cell conoids represent only about 20% of the total population) then replace the term « mis-localization » as the protein is rather no longer present than found at another localization within the parasite.

Discussion

Can the authors comment on why ASAF1 depletion lead to the absence of IMC formation in nascent daughter cells? Is that an indirect effect linked to the loss of apical polarity in nascent daughter cells or do ASAF1 could play a more direct role on IMC formation ?

Reviewer #3

(Remarks to the Author)

Wei Li et al in the manuscript “Novel apical ring protein essential for conoid complex assembly and daughter cell formation in *Toxoplasma gondii*” have applied a variety of methods with a focus on imaging and showed that that previously identified conoid gliding protein is crucial for targeting actin nucleator formin 1 and other preconoidal ring proteins to preconoidal rings of *Toxoplasma gondii*. They applied cryo-correlative light and electron microscopy as well as cryo-electron tomography to study conoid gliding protein KO parasites, which convincingly revealed an absence of rhoptries (preconoidal ring). Overall, the manuscript is mainly dedicated to the parasitology community, however, the cryo-CLEM method applied is very useful and can be adapted to a variety of projects. It is well-written and presents high-quality imaging data using state-of-the-art imaging techniques. Several minor issues should be addressed in particular to provide statistics on cryo-ET data and to clarify and better present the data for a broader audience. I have mainly focused on assessing the cryo-EM part of the manuscript.

1) Please clarify and consolidate the nomenclature and terms used throughout of the manuscript. For example, rhoptries are not indicated in the legend of Figure 3b and it is not clear if preconoidal ring and rhoptries refer to the same thing.

2) While tomogram data are of high quality, only overview slices are shown. It would be beneficial to show zoom-in of the rhoptries. Also, the color code chosen in the segmentation and segmentation legend (Figure 3a) are not easy to follow as some of the colors are similar. Also indicate that each row represents the same tomogram, perhaps use labelling – bottom slice, central slice, top slice of the tomogram.

3) Line 13, line 38, page 7: Please provide the exact numbers of tomograms which were analyzed and report the frequency of the observed events:

- “While analysing the cryo-ET data, we noticed that the apical plasma membrane was often disrupted during this process.”
- “However, in 33% of cgp iKO parasites, rhoptries could not be observed within the tomogram field of view (Fig. 3b and Extended Data Fig. 2; Supplementary Video 4).”

4) Do all wild-type parasites contain preconoidal ring? Does the absence of preconoidal ring has any influence of overall shape and size of the conoid?

5) It would be beneficial to introduce and cite previous cryo-ET studies on *Toxoplasma* apical ring. For example: <https://doi.org/10.1073/pnas.2111661119>; <https://doi.org/10.1038/s41467-023-37327-w>

6) Throughout the manuscript, cryo-EM/ET abbreviations could be better used. Sometimes authors use cryoET, cryo-ET, cryo-electron microscopy instead of cryo-EM etc.

7) Methods Title:

“Ultrastructural TEM” should be changed to Thin-section TEM of Epoxy-embedded samples

“Cryo-correlated light and electron microscopy images alignment” should be changed to “Correlation of cryo-light and electron microscopy data”

8) Cryo-ET data acquisition (method section): “fluence” is not commonly used and it should be replaced by electron dose rate in e-/Angstrom²/sec. Total dose should be reported in e-/Angstrom².

9) Please provide info if any motion correction procedure was applied during tomogram reconstruction.

10) What was used as correlation markers for cryo-CLEM, please specify the features that were used, this might not be obvious to a broad audience.

Reviewer #4

(Remarks to the Author)

In this paper, Li et al, study the role of a previously identified protein CGP in conoid formation and find that it is important for the stability of the preconoidal rings in mature conoids. Using proximity-based labelling approaches, they then identify 15 other candidates that are likely involved in the formation of the conoid and go on to characterise the most promising candidate, which they term ASAF1. This protein appears to be involved in early conoid assembly and hence SPMT localisation and daughter cell formation.

This is a nice story and will be of interest to the parasite community. The conoid and SPMTs are an exciting topic at the moment with many groups studying them and this paper brings us closer to understanding how these important and complex structures are assembled, organised and stabilised. The data is solid, clear and uses an impressive array of imaging techniques that result in beautiful figures. Some parts could be improved by providing more numerical data (eg. how many cells imaged and of those how many had X phenotype) and the wording of some speculations could be toned down in places.

I only have minor comments:

Abstract and Introduction

- Line 1 page 2: “The conoid complex is a unique structure in apicomplexan parasites”

Is the conoid unique to apicomplexan parasites? I think it is also found in some other alveolates.

- Line 3 and 4 page 2: “It is a dynamic organelle that plays an important role for initiation of gliding motility” and Line 14 and 15 page 3 “From the APRs the minus end of 22 subpellicular MTs radiate and extend to approximately two-thirds of the parasite’s length”

The abstract and introduction are written as if this is about all apicomplexa – however the details are Toxoplasma specific. Eg. Most Plasmodium motile stages don't have a conoid and they don't have 22 microtubules. Either make the abstract and introduction just about Toxoplasma or make it more general and then remove specific statements like these (and add references for other apicomplexa).

Results and Discussion:

- Lines 28 and 29 on page 7: "However, in 33% of cgp iKO parasites, rhoptries could not be observed within the tomogram field of view".

Can you clarify this – so in 67% you can see proper localisation of rhoptries? And in what percentage of WT can you see rhoptries within the tomogram field of view?

If you see proper localisation in 67% of iKO cells and not in 100% of WT cell (for example) then I don't think the lines 2 and 3 on pg 6 "our findings highlight the importance of PCRs in ensuring the proper attachment of rhoptries to the apical complex." can be stated.

Please also add how many tomograms were collected and analysed of WT and iKO to give us a better understanding of the significance of 33%.

- Line 18 page 11: "Subsequent analyses in regular imaging"

What is "regular imaging"?

- For the TEM analysis page 11 – please give numbers – how many WT and ASAF1 KO cells were imaged, for both how many cells did you see daughter conoids or correct IMC formation. Please also have a supplemental figure showing more cells than the one for each WT and KO in figure 5b.
- Lines 16 and 17 on page 14: "This appeared especially important, since our analysis suggests that contact of mature PCRs with the plasma membrane is required for its stability."

I don't understand where you got this from – what analysis have you done to suggest this? This isn't clear to me and should be removed (unless you have the analysis to show this).

- Line 22 on page 14 – ICMTs is used for the first time and not defined.
- Lines 2,3,4 on page 15: "We therefore speculate that contact of the PCRs to the plasma membrane is required for final maturation and stabilisation of the PCRs, which then requires CGP, Pcr4 (and probably Pcr5)."

Again, as above, I don't understand where this speculation has come from. Here it is called a speculation and above it is called "our analysis"

- Figure 4b. Why does the scale change in images 1-6. It would make it much easier to understand what was going on if this was kept consistent.
- Figure 6e. The paper sometimes feels like multiple stories and what I am missing is something that links all the proteins analysed in this paper together in a clear way. I would at least include CGP in this model figure.
- Some of the discussion reads like a repeat of the results, this could be made more succinct.

Materials and methods:

- Ultrastructural TEM – missing a few pieces of information like pixel size.
- Tomogram reconstruction – again missing information, what software was used for frame alignment (assuming you collected frames – this is not mentioned in cryo-electron tomography), CTF correction etc.

Version 1:

Reviewer comments:

Reviewer #1

(Remarks to the Author)

I appreciate the effort that the authors have taken to revise the manuscript. They have addressed my comments thoroughly, and in cases where changes were not made (STED vs U-ExM), the provided justifications were clear and reasonable. I find

the revised version of the manuscript significantly improved and recommend it for publication.

Thank you to all the authors for this excellent study that the field will certainly appreciate for its technical excellence and interesting cell biology.

Reviewer #2

(Remarks to the Author)

I thank the authors to have performed the additional experiments that were asked and to have greatly improved the manuscript structure. I have no further concerns.

The manuscript is of high quality, suitable for publication.

Reviewer #3

(Remarks to the Author)

All my comments and suggestions were addressed and the figure showing Cryo-ET data was improved.

Reviewer #4

(Remarks to the Author)

The revised manuscript by Li et al is much improved and clearer than the first version. I am satisfied with the changes made in response to my comments.

Please can the authors just add details of how segmentation of cryo-tomograms was carried out in the final manuscript.

Otherwise I am happy for the manuscript to be published.

General Comments

We thank all reviewers for their thoughtful feedback, which greatly helped to refine and improve the manuscript. In response, we implemented several global revisions to enhance clarity, cohesion, and data interpretation:

- **Strengthened narrative structure:** We clarified the temporal progression of conoid complex assembly, positioning CGP as a stabilizing factor in mature parasites and ASAF1 as an essential early assembly factor in daughter cells. This logical framework is now consistently articulated across the Abstract, Introduction, Results, and Discussion.
- **Consistent terminology and structural definitions:** We revised the manuscript to use precise and standardized terms throughout. Structures are now referred to as “preconoidal rings (PCRs),” “apical polar ring (APR),” or “conoid,” as appropriate. Ambiguous terms like “apical rings” were removed. We also clarified that CGP is a PCR-associated protein and acknowledged in the Discussion that its name—originally based on early localization assumptions—does not fully reflect its refined structural assignment.

- **Use of STED versus Expansion Microscopy (ExM):**

As all reviewers raised the topic of imaging approaches, we would like to address this directly. While Expansion Microscopy offers excellent resolution and volumetric context—particularly valuable for large-scale structural visualization—it involves extensive sample processing steps such as gelation and denaturation. These procedures can, in some instances, introduce structural distortions or mislocalization artifacts. In contrast, STED microscopy allows high-resolution imaging of native cellular structures with minimal sample manipulation. For this reason, our laboratory routinely employs STED as the **primary super-resolution method**, especially for **colocalization analyses and structural assignments**. In fact, we rely on STED to validate the robustness and reliability of ExM-based observations to exclude possible artifacts.

While we appreciate and understand the suggestion to use ExM for additional analyses, **in our experience, STED consistently provides reliable and reproducible colocalization data**. As such, we do not believe that every experiment necessitates ExM, particularly when STED already yields high-confidence insights. That said, in this study, we have used ExM selectively in cases where it offered specific value or additional resolution beyond what STED could deliver.

Please find our detailed response in blue

REVIEWER COMMENTS

Reviewer #1 (Remarks to the Author):

Background:

CGP is a crucial protein of the preconoidal rings, initially identified in a splitCas9 phenotypic screen by the Meissner/Jimenez-Ruiz group (Li, Grech, Stortz et al., Nat Micro. 2022). At that time, CGP was shown to play a role in actin-based motility, significantly affecting invasion and egress. Around the same period, the Soldati-Favre group characterised various preconoidal rings proteins, including the

essential motility factor Formin1 (FRM1), which was observed to localise to the pre-conoidal rings (Dos Santos et al., Nat Micro. 2022). Notably, certain mutants of pre-conoidal rings proteins, specifically Pcr4 and Pcr5, have been found to be essential for maintaining the stability of the pre-conoidal rings.

In a separate development, recent reports from various teams have provided a detailed account of the early stages of daughter cell formation, using advanced high-resolution microscopy techniques (Arias Padilla et al., MBoC, 2023 ; Arias-Padilla, Munera Lopez et al. JCS. 2024; Engelberg et al., Biorxiv, 2024 ; Haase et al., MBoC, 2024). Some of those reports finally reported on the initial role of gamma-tubulin in the early development of daughter cells scaffolds. While gamma-tubulin is crucial for microtubule polymerisation in other organisms, it has never been observed at the apical polar ring, which is considered the microtubule organising centre (MTOC) in apicomplexan parasites.

Findings:

In this study by Li, Koczy et al., the role of CGP is examined in detail. The authors demonstrate that CGP is essential for maintaining the stability of the pre-conoidal rings in mature parasites, though APH not in daughter cells. The absence of CGP mirrors the effects of Pcr4/5 depletion, resulting in the specific loss of pre-conoidal ring proteins. Consequently, the severe motility, invasion, and egress phenotypes observed in previous studies can now be attributed to the disruption of the pre-conoidal rings and loss of FRM1.

In the second part of the manuscript, the authors conduct a proximity labelling experiment to identify new conoidal complex proteins, using CGP and FRM1 as markers. While previously known proteins of the complex are identified in both CGP and FRM1 BioID assays, other, as yet undescribed proteins are also detected. The authors particularly highlight ASAF1, an essential protein located at the apical polar ring in developing daughter cells. Depletion of ASAF1 results in a marked defect in daughter cell formation, though it does not impact the formation of key structures such as centrosomes, kinetochores, and the striated fibre. Finally, the authors demonstrate that ASAF1 depletion disrupts the assembly of the conoid complex and subpellicular microtubule scaffold, thereby abrogating daughter cell formation.

General comments:

This study is of high quality, presenting an impressive volume of data and a broad range of methods. However, while the depth of data is remarkable, the manuscript's narrative could be more cohesive. It currently features two main threads: firstly, the continuation of the CGP characterisation begun in the authors' previous study (Li, Grech, Stortz et al., Nat Micro. 2022), and secondly, the characterisation of ASAF1, a protein involved in daughter cell formation.

We condensed the CGP characterization section and moved redundant data to supplementary material. STED and live-cell imaging now support only essential claims (Fig. 2, Supplementary Video 1). Cryo-ET is prioritized early (Fig. 3).

The first part, covering the characterisation of CGP, feels somewhat lengthy and complex, spanning three main figures when it might be more effectively conveyed in one or two. Presenting the cryo-ET data illustrating the loss of the pre-conoidal rings in the absence of CGP as the initial figure could make the current Figure 1 redundant, possibly placing it better as supplementary material. Moreover, since the authors already established in their prior publication that CGP is indeed a resident protein of the pre-conoidal rings, proving this point again here is unnecessary.

We would like to thank the reviewer for their thorough and insightful comments, which will undoubtedly improve the manuscript's clarity and impact. In our revision we condensed the CGP characterization section and moved redundant data to supplementary material.

STED and live-cell imaging now support only essential claims. We have reorganized and combined data from previous Figures 1 and 2 into a single figure (revised Figure 1), prioritizing the most essential findings. Cryo-ET is prioritized early (Fig. 2).

We believe that these changes make the CGP characterization section more concise and focused, while preserving critical information. Regarding the inclusion of colocalization data: While our previous publication (Li et al., 2022) suggested CGP is a preconoidal ring protein, the colocalization of CGP with other preconoidal proteins (e.g., Pcr4) validates this assumption.

We hope that these changes adequately address the reviewer's concerns and improve the manuscript's readability and focus.

I find the second part of the manuscript, focusing on ASAF1 and its role in daughter cell formation, particularly engaging. These findings are also well-timed, aligning with recent publications. The super-resolution microscopy, expansion microscopy, and video microscopy provide compelling insights into daughter cell formation. However, the author should be more careful about the use of the terms 'conoid' instead of 'conoid complex' or 'APR' which sometimes makes some conclusions difficult to understand.

We revised the terminology employed throughout the manuscript to be more consistent and accurate (see general comments)

Specific comments (in order of appearance within the text):

- In the text, it would be preferable if the author could avoid the term "apical rings," as this might cause confusion for readers who are not specialists in the field. The terms "apical polar rings" and "preconoidal rings" refer to specific, well-defined structures and should ideally be used as such. Along the same lines, the term "conoidal" may benefit from more precise usage. It is certainly appropriate to describe a protein as "conoidal" if it localises to the conoid. However, it may be more accurate to refer to a protein within the broader apical complex as an "apical complex protein," rather than "conoidal." This is especially relevant to the naming of CGP (Conoid Gliding Protein), which, in my view, could benefit from reconsideration, as it is not a conoid protein but a preconoidal rings protein and is involved in functions beyond gliding, which may lead to some ambiguity.

We agree that consistent and precise usage of terms is essential to ensure clarity for readers. As suggested, we have carefully revised the manuscript. Terms like "apical rings" and "conoidal protein" were replaced with "APR" and "PCR protein."

We also thank the reviewer for the thoughtful suggestion to reconsider the naming of "CGP" (Conoid Gliding Protein) for improved clarity. While we acknowledge that the name may no longer fully reflect the protein's refined localization and multifunctional role, CGP was originally identified in our previous study (Li et al., 2022) as a conoid complex-associated factor involved in gliding motility. To maintain continuity with the literature and ensure traceability across publications, we have retained the original nomenclature. However, in light of the broader functional insights presented in this manuscript, we have revised the text to clearly state that CGP is a preconoidal ring protein with functions that extend beyond motility, thereby minimizing potential confusion. More broadly, we agree that a systematic revision of nomenclature for PCR-associated proteins may be warranted, and we believe this could be most effectively addressed in a future review.

- In the introduction, particularly in the first and second paragraphs, it would be helpful for the authors to more clearly distinguish between aspects specific to Apicomplexa in general and those specific to Toxoplasma. For instance, while it is accurate that the conoid complex contains 'unique secretory organelles' across all Apicomplexa, it should be noted that Plasmodium and Cryptosporidium do not possess ICMTs within the conoid. Rephrasing these sections to clarify what constitutes general information about Apicomplexa and what is specific to Toxoplasma would enhance the reader's understanding of the manuscript's particular focus.

We agree that distinguishing between features conserved across Apicomplexa and those specific to *T. gondii* can improve clarity. However, as our study focuses exclusively on *T. gondii*, and we do not present comparative data from other apicomplexans, we have chosen to revise the introduction to describe *T. gondii*-related features. We believe in this way, it avoids overgeneralization and keeps the text aligned with the scope of our study.

- In the introduction, it may be helpful to specifically outline the proposed model for the formation of the SPMTs basket as described by Engelberg and colleagues. While this model is explained later in the text, providing a clear summary in the introduction could improve reader comprehension. I am specifically referring to the initial formation of microtubules near the centrioles, which are then subsequently incorporated into the developing APR.

A summary about formation of the SPMTs basket were added in the introduction part. Now it reads:

“Early on, daughter cell construction initiates near the duplicated centrioles. Nascent SPMTs emerge as discrete “rafts,” each composed of paired microtubules anchored to the APR, while the conoid and components of the IMC begin to form in parallel. These structures extend toward the centrioles, and eventually, the APR and conoid assemble into a complete ring structure. During this process, it is proposed that SPMTs are nucleated near the centrioles and subsequently incorporated into the forming APR. Ultimately, five distinct SPMT rafts, each containing four to six microtubules, are established, resulting in the nucleation of all 22 SPMTs. The conoid complex, SPMTs, and IMC then extend in a basal direction, until the daughter cells mature and bud from the mother cell “. (Page 4, line 4-12)

- Page 3 line 19-20: rephrase “with the unconventional myosin MyoH localized at the conoid” instead of “apical localised unconventional myosin MyoH” for clarity.

We have implemented this change.

- page 5 line 1: all abbreviations are already defined in the introduction and can be used without writing the name of the proteins GAC, AKMT and MyoH in full.

We have implemented this change.

- page 5 line 4: It would be helpful if the author could provide a bit more explanation about the Halo tag, including what it is and why it has been utilised in this study.

We thank the reviewer for this helpful suggestion. To provide clarity without disrupting the flow of the main text, we have included a brief explanation of the HaloTag system in the Materials and Methods section (see page 19, lines 11-14).

“The HaloTag system was utilized to label CGP and other proteins due to its versatility in conjugating different fluorophores and its ability to provide high signal-to-noise ratios and specificity when combined with Janelia dyes (Promega). This approach was particularly advantageous for imaging low-expressing proteins”

- Figure 1B: The observation that MyoH is unaffected in the absence of CGP appears somewhat redundant, given that the previous study already demonstrated that neither apical polar ring (APR) nor conoid proteins were impacted by CGP depletion.

In agreement with the comment, we have moved the MyoH panel from previous Figure 1b to the supplementary data (Supplementary Fig. 1). While we recognize that previous studies demonstrated that CGP depletion does not impact the apical polar ring (APR) or conoid proteins, our focus in this section was on motility factors associated with the conoid complex. Thus, examining MyoH alongside other key motility-associated proteins was relevant to establish a comprehensive understanding of CGP's role in motility initiation. Our results indicate that, among these factors, only FRM1 is affected by CGP depletion, further highlighting its unique dependence on CGP.

- Figure 1B-C-D: Could the author clarify why certain SYFP2-tagged proteins are detected directly via SYFP2 fluorescence (MyoH / GAC), while others require detection with an anti-GFP antibody (AKMT)?

The use of antibodies in certain cases was determined by the natural expression levels of the tagged proteins. For proteins with sufficient expression levels, SYFP2 fluorescence was detectable without additional enhancement. However, for proteins with lower expression levels, antibody detection was employed to improve the signal-to-noise ratio for imaging purposes. To clarify this point, we have added a sentence to the materials and methods section (see page 23 lines 14-18).

“For the detection of SYFP2-tagged proteins, the choice between direct visualization via SYFP2 fluorescence and the use of anti-GFP antibodies was based on the natural expression levels of each protein. Proteins with sufficiently high expression levels were visualized directly, while lower expression proteins were detected with antibodies to enhance the signal-to-noise ratio for improved imaging clarity. This was indicated in the figure legends as needed.”

- Figure 2A: I am not entirely convinced by the STED images throughout the manuscript. For instance, in Figure 2A, it's unclear how much additional resolution STED provides over the confocal imaging. Given the authors' evident proficiency with expansion microscopy and the significantly higher gain in resolution it offers, I believe it would be beneficial to use expansion microscopy in place of STED. Additionally, as previously mentioned, demonstrating that CGP is a preconoidal ring protein feels somewhat redundant, as this was already convincingly shown in the previous study, where the co-localisation with SAS6L/RNG2 was particularly clear.

See also our general comments above. While Expansion Microscopy (ExM) provides volumetric context and high resolution, it involves extensive sample processing that can introduce structural artifacts. In contrast, STED allows imaging of native structures with minimal manipulation. Our lab routinely uses STED as a primary validation tool for colocalization, especially for apical complex proteins, to ensure imaging fidelity. Therefore, we do not believe every analysis requires ExM when STED already provides robust colocalization. Furthermore, to support imaging of low-abundance proteins like CGP, we used the HaloTag system in combination with Janelia Fluor dyes. This approach offers high signal-to-noise and flexible labeling, making it ideal for super-resolution microscopy. A brief explanation of the HaloTag system and its advantages has been added to the Methods section (see above). Where appropriate, we complemented STED with ExM when it provided added structural insight.

Regarding the localization of CGP to the PCRs, our previous studies suggested CGP's presence in this region based on co-localization with SAS6L and RNG2. In this manuscript, we directly visualize CGP's position relative to PCR proteins, providing a more straightforward and complementary validation of our earlier findings. We have now moved the co-localization figure to the supplementary figures.

- Page 5 line 24: Could the authors please clarify what is meant by the term 'second apical polar ring'? Are there multiple apical polar rings present and therefore several MTOCs? What are the several APRs described throughout the text?

We thank the reviewer for pointing this out and apologize for any confusion. There is indeed only one apical polar ring (APR); however, recent cryo-ET studies have revealed that it consists of three spatially distinct layers or subdomains. These layers are part of the same structural unit and collectively define a single APR, which serves as the microtubule-organizing center (MTOC) at the apical end of both the inner membrane complex (IMC) and the subpellicular microtubules (SPMTs). To prevent confusion, we have revised the manuscript to consistently refer to this as “the APR” and clarified that it represents a single MTOC.

- Figure 2: Given the author’s expertise in using fluorescent markers and video microscopy, it would be advantageous to conduct video microscopy on preconoidal ring-tagged lines when CGP is downregulated. This would provide critical data regarding the timing of preconoidal ring loss during daughter cell hatching. Capturing video footage with images taken every 5 to 10 minutes and spanning a full cell cycle would yield highly compelling results. It is unfortunate that not many researchers in the field are utilising video microscopy, and I salute the author for their efforts in this area.

We thank the reviewer for the kind words and for highlighting the value of video microscopy in studying dynamic cellular processes. We fully agree that time-lapse imaging of preconoidal ring dynamics during daughter cell hatching would be highly informative. We did attempt such experiments; however, most PCR-associated proteins investigated in this study are expressed at relatively low levels, making them insufficiently bright for reliable live-cell imaging across extended timeframes. Moreover, acquiring full z-stacks every 5–10 minutes over an entire cell cycle introduces challenges with photobleaching and positional drift, increasing the risk of losing focus on the apical structures. To address this limitation, we instead acquired still images at representative stages of the cell cycle, which together reconstruct the temporal sequence of PCR assembly and disassembly.

- Page 6 line 19-21: The sentence “We speculate that this stabilisation step requires the contact of PCR proteins with the plasma membrane which only occurs after hatching of the daughter cells from the mother” would be more appropriately placed in the discussion section, as there is currently no supporting evidence.

We removed it from the results and mention it accordingly in the discussion section..

- Page 7 line 21-23: Could the authors explain why some parasites are still able to carry out conoid protrusion even in the absence of preconoidal rings? This observation appears to challenge the current model regarding conoid protrusion and the role of actin in this process. Additionally, what is meant by the phrase 'in some cases'? Providing quantification for these observations would be extremely beneficial.

We thank the reviewer for this suggestion. To clarify and quantify this phenomenon, we performed Expansion Microscopy (ExM) using tubulin staining to assess conoid protrusion in CGP-depleted parasites. We found that 5.6% (25/426) of CGP-depleted parasites were able to extrude the conoid, consistent with previous findings of 5.45% (33/605) in Pcr4-depleted parasites (Dos Santos Pacheco et al., 2022). Additionally, in our cryo-ET dataset, 21.7% (5/23) of CGP-depleted parasites and 80% (12/15) of wild-type parasites with intact apical membranes showed either full or partial conoid protrusion.

These results demonstrate that although PCRs—and thus apical actin polymerization—are critical for efficient conoid extrusion, a small subset of parasites can still protrude the conoid in their absence. This suggests that additional structural elements may partially compensate or contribute to the

mechanical basis of protrusion. We have clarified this point and included quantification in the revised manuscript.

- Figure 3: I am unable to locate the number of parasites reconstructed for the cryo-electron tomography under each condition. It would be beneficial for the authors to include quantification to support their conclusions regarding the integrity of the ICMTs, APR, and MVs.

For our ultrastructural analyses, we only considered intact parasites with membranes visible around their apical region. This approach allowed us to avoid that potentially altered phenotypes from disrupted parasites could influence the assessment of the presence of preconoidal rings and the description of other organelles.

During the process of revising our manuscript we analyzed 15 additional parasites. Overall, we analyzed 38 parasites from two separate biological experiments and found that 26.09% (6 parasites out of 23) cgp iKO parasites did not have the rhoptries docked in the conoid in the observed field of view. For the WT parasites, 13.33% (2 out of 15 analyzed) were lacking the rhoptries docked in the observed field of view.

- Page 7 (line 28-29) and page 8 (line 2-3): Was the disruption of the rhoptries anticipated in the absence of preconoidal rings? This appears to be a rather surprising finding. Is the apical vesicle (AV) (Mageswaran et al. Nat Comm. 2021; Segev-Zarko et al. PNAS Nexus. 2022) still present in absence of the PCRs? If this observation holds true, it would be beneficial for the authors to strengthen their data by employing additional techniques that are more suited for quantification. Furthermore, conducting a rhoptry secretion assay would be valuable to demonstrate how the loss of preconoidal rings impacts invasion, in addition to its effects on motility.

We did not anticipate disruption of rhoptries or rhoptries docking in the absence of preconoidal rings. We do not observe other structures, like the microtubules associated vesicles (MVs) or intraconoidal microtubules (ICMTs) to be affected. We observed that 21.7% of CGP-depleted parasites lacked the apical vesicle (AV), while no such defect was detected in wild-type parasites.

In the conducted rhoptry secretory assay we observe a reduction in the rhoptry secretion of CGP-depleted parasites compared to wild-type, strengthening the evidence for the phenomena observed in our cryo-ET data.

- Page 8 line 18-19: I find the phrase 'known or predicted localization at the apical region or unknown localizations predicted by HyperLOPIT' a bit unclear. Could the authors clarify the criteria they used for this classification? As it currently reads, it seems that 'predicted apical localisation' or 'unknown localisation' could refer to almost any location within the cell.

We have clarified the criteria used to define localization in our protein selection process. It reads now:

“To narrow down the list of candidate proteins, we applied the following criteria: 1) significant enrichment (fold change over 4.5), 2) a phenotypic score below -1, indicating an important function during the asexual cycle of the parasite or known to be essential, and 3) either known localization at the conoid complex based on published localization studies, or predicted apical localization with a final probability score of 1 or no assigned localization (N/A) or any predicted localization (regardless of apical or not) with a final probability score below 1, predicted by HyperLOPIT” (Page 9, line 6-12).

Only 15 candidate genes meet all the criteria.

- Page 8 line 21: Please state clearly which proteins couldn't be tagged. Alternatively, this information can be added to the supplementary table 1.

As suggested, we added the information about the candidate proteins in Supplementary Table 1.

- Extended data Fig.5F-G-H: Do these data suggest that ICAP16 is a resident protein of the PCRs? It might be more straightforward to confirm this through expansion microscopy.

We thank the reviewer for this suggestion. The PCR localization of ICAP16 has been previously demonstrated using 3D-SIM microscopy (Koreny et al., 2021), and our current data show that ICAP16 signal is lost from the apical region in mature parasites upon CGP depletion. Taken together, we believe this provides strong support for ICAP16 being a resident PCR protein. While expansion microscopy could offer an additional layer of spatial resolution, we consider the existing evidence sufficient to support this conclusion in the context of the present study.

- Extended data Fig.6A-B-C-D: Do these data suggest that DAP1 is a resident protein of the PCRs? It might be more straightforward to confirm this through expansion microscopy.

The PCR localization of Dap1 was previously demonstrated by 3D-SIM microscopy (Koreny et al., 2021). Together with our data (CGP depletion resulted in Dap1 were absent in the apical region, we believe that Dap1 is a resident protein of the PCRs.

- Extended data Fig.7A-B-C-D: I suggest doing expansion microscopy instead of STED to prove the PCRs localisation of PCKMT.

See our response regarding co-localisation analysis using STED vs ExM above.

- Page 9 line 18-20: A straightforward approach, such as expansion microscopy, to localise PCKMT, DAP16, and ICAP16 would provide clear confirmation of their positioning at the PCRs. This would also allow for more certainty in describing these proteins as being located at the PCRs, rather than stating that they 'appear to be localised' there.

We thank the reviewer for suggesting expansion microscopy as a useful method to confirm PCR localization. However, as discussed above, our lab routinely uses STED as a primary validation tool for colocalization, especially for apical complex proteins, to ensure imaging fidelity. Therefore, we do not believe every analysis requires ExM when STED already provides robust colocalization. Furthermore, our existing data, together with previously published studies, provide robust evidence that PCKMT, Dap1, and ICAP16 are PCR proteins. Specifically, 3D structured illumination microscopy (3D-SIM) (another alternative for super-resolution imaging) has already unambiguously localized ICAP16 and DAP16 to the PCRs (Koreny et al., 2021). Moreover, CGP depletion leads to complete loss of PCR structures and the concomitant disappearance of PCKMT, Dap1, and ICAP16 from the parasite's apical complex while the localisation of conoid proteins such as SAS6L (Li et al., 2022) and MyoH (this study), and APR protein RNG2 (Li et al., 2022) were not affected. Taken together, these findings justify our description of PCKMT, Dap1, and ICAP16 as resident PCR proteins without additional expansion microscopy data. We now have also removed "appear to" from the corresponding text for clarity.

- Page 9 line 19: Maybe replace "from the conoid" by "from the conoid complex/apical complex/apical pole of the parasite" to avoid confusion.

We now replaced "from the conoid" with "from the apical complex" as suggested.

- Extended data Fig.8B: It would be helpful to clarify that TGGT1_238170 refers to IAP3, as this is not indicated in the text, table, or in the graph in panel A. This omission, in addition to the high amount of data and proteins investigated, makes it a bit challenging to follow the sequence of panels describing the numerous BioID candidates.

We have now clarified in the main text that TGGT1_238170 refers to IMC-associated protein 3 (IAP3). It reads:

“Localisation analysis demonstrated that TGGT1_238170, known as IMC-associated protein 3 (IAP3), is indeed cell cycle dependent expressed and displayed dynamic localization patterns during replication (Extended Data Fig. 9b), consistent with a recent study”. (Page 10, line 26-29)

- Extended data Fig.8C: It would be helpful if the authors could provide an explanation of what TLP1/TLAP1 refers to, at least in the figure legend, as it is currently not described in the main text, figure legends, or supplementary figures. Additionally, primers used for tagging the gene do not appear to be specified. The dark blue signal shown by IFA is somewhat unclear in the images presented. If this refers to the TLAP1 protein identified by Ke Hu’s team (Liu et al., Eukaryotic Cell, 2013), it would be useful to state this clearly and to reference their study.

We have now clarified that TLAP1 refers to the TrxL1-associating proteins by including this information directly in the figure legend for Supplementary Fig. 9c. We have also added the corresponding reference as suggested. Additionally, the primers used for TLAP1 tagging have been included in Supplementary Table 2 and 3. To improve clarity, we have added an extra panel in the figure to better illustrate the TLAP1 signal and indicate the corresponding parasite stages.

- Extended data Fig.8F: Here, it may be more straightforward and precise to use expansion microscopy to co-localise ASAF1 with CGP, rather than relying on the current images. Additionally, it could be worthwhile to explore co-localisation with Pcr7 (Dos Santos et al., Nat Micro. 2022), which, to my knowledge, is the only protein described specifically at the pre-conoidal rings of daughter cells.

We thank the reviewer for these thoughtful suggestions. Our primary aim in Extended Data Fig. 8F was to illustrate the cell cycle-dependent expression of ASAF1 and its temporal co-localization with CGP during daughter cell formation. We believe the confocal images effectively convey this relationship and serve their intended purpose within the context of this figure. To further clarify ASAF1’s precise subcellular localization, we complemented this with STED microscopy using markers of specific conoid complex substructures (Fig. 3a) and Expansion Microscopy (Fig. 3b), which revealed that ASAF1 localizes to the apical polar ring (APR). While co-localization with Pcr7 could be of interest in future studies, we consider the current dataset sufficient to define the spatiotemporal dynamics of ASAF1 in the context of this study.

- Page 10 line 19: RNG2 is not a marker of the conoid but a marker of the APR.

Now we changed ‘conoid’ to ‘conoid complex’. (page 11, line 15)

- page 11 line 7-11: I have some reservations regarding the conclusions drawn here, particularly the statement that ‘formation of the conoid and consequently for recruitment of additional APR proteins’. I am not entirely convinced that conoid formation is a prerequisite for the assembly of the APR. It seems plausible that the APR may actually form first, followed by the SPMTs and then the conoid. Additional experiments would be beneficial to support such a conclusion, as this would be particularly important for our understanding of the sequence in which the sub-compartments of the conoid complex are assembled. Furthermore, the claim that ASAF1 is essential for gamma-tubulin complex recruitment is not supported by the data presented here. It would be helpful if the authors could assess gamma-tubulin levels and localisation in the ASAF1-null mutant, and vice versa, to substantiate this point.

We thank the reviewer for these important observations. We agree that the apical polar ring (APR) is likely assembled prior to the conoid and subpellicular microtubules, and we have revised the relevant text to clarify this sequence of events (see page 12, line 10). This interpretation is supported by our

own data showing that ASAF1 appears before detectable conoid or microtubule structures, as well as by recent findings from Padilla et al. (2024), who demonstrated that APR2 is detectable earlier than both the conoid and nascent microtubules.

In response to the reviewer's concern about γ -tubulin, we now provide additional data showing that, upon ASAF1 depletion, γ -tubulin remains detectable at the spindle poles and around the centrioles, but the signal typically associated with the APR region in wild-type parasites is absent (see Fig. 5b). This suggests that while ASAF1 is not required for general γ -tubulin expression or localization to centrosomes, it may be critical for γ -tubulin recruitment or retention at the APR, likely due to the disrupted architecture of the conoid complex in ASAF1-null mutants. We have clarified this interpretation in the revised text.

- page 11 line 26-27: I am not sure whether microtubules extending from the centrosomes, without any attachment to structures resembling an APR, should be referred to as 'subpellicular' microtubules.

We have removed the term subpellicular from this sentence.

- page 11 line 28-29: The APR and conoid are two different structures. The author should remove the conoid mention from the sentence.

We thank the reviewer for pointing this out. The sentence has been revised accordingly. It now reads:

"This is in good agreement with a role of ASAF1 for formation of the APR that serve as MTOCs for daughter bud formation (Fig.4b)." (Page 13, line 1-2)

- Figure 5C: Do the author have any hypothesis/explanations regarding the 'membrane folds' observed in the ASAF1-null mutant?

We hypothesize that these membrane folds appear as a "normal" expansion of the plasma membrane to meet the needs of emerging daughter cells, which, in this case, fail to form properly. This suggests that while other processes associated with cell division are still occurring, the correct assembly of daughter cells—and consequently the completion of cytokinesis—is disrupted.

- page 12 line 25: "ASAF1 is critical for formation of the conoid complex" or "ASAF1 is critical for formation of the APR" would be more precise.

As suggested, we have changed 'conoid complex' to 'APR' as suggested for accuracy.

- Figure 6A: Do the authors have any insight into the specific location within the conoid complex where SFA2 is anchored?

At this stage, we can only speculate that SFA2 is anchored to the APRs and that the failure to form these structures results in the disengagement of SFA2 from the conoidal/APR-associated protein clusters. Further studies are required to address this interesting question.

- Extended data Fig.10H-I: I'm a bit puzzled as to why the two graphs use different metrics. One graph examines dividing versus non-dividing, while the other focuses on the number of puncta for each protein. Could you help clarify the reasoning behind this difference? I think they should both be assessing the number of abnormal cytoplasmic puncta for each protein. It might have been interesting to also add Centrin1 to the list of protein of interest in this experiment as it has been shown to be fragmenting in gamma-tubulin depleted parasites.

We thank the reviewer for raising this important point. We initially attempted to use a uniform metric—specifically, quantifying the number of cytoplasmic puncta per nucleus—for all proteins following ASAF1 depletion. However, this approach proved unreliable due to the tendency of several

markers to form dense or overlapping clusters. In such cases, it was difficult to distinguish whether a signal represented a single focus or multiple discrete puncta, particularly in vacuoles containing large aggregates. To address this limitation, we adapted the analysis method for each marker in a way that best captured the observed phenotype. We agree this results in different readouts, but believe the chosen approach more accurately reflects the protein-specific patterns of disruption.

Regarding Centrin1, we thank the reviewer for the suggestion. We examined Centrin1 separately by IFA (see Supplementary Fig. 11a) and consistently observed its presence in ASAF1-depleted parasites. Moreover, centriole formation appeared largely intact, particularly during early stages of division, as also observed in our expansion microscopy images. These results suggest that ASAF1 depletion does not significantly affect Centrin1 localization or centriole integrity, in contrast to what has been described for γ -tubulin mutants.

- Figure 6B-D: Perhaps the authors might consider conducting a western blot analysis to assess the stability of the various proteins in the ASAF1-null mutant. This could help clarify the statement, "in the case they are still expressed" (page 13 line 14-15).

We appreciate the reviewer's suggestion and fully agree that assessing protein stability in the ASAF1-null background would be valuable. We attempted to perform Western blot analyses for several conoid complex proteins following ASAF1 depletion. However, these experiments proved technically challenging, because parasites rapidly die intracellularly upon ASAF1 deletion. Therefore, it was difficult to obtain sufficient parasite material without significant contamination from host cell proteins. Additionally, many of the candidate proteins—such as CGP (~500 kDa) and FRM1 (~400 kDa)—are exceptionally large, further complicating their detection by standard Western blotting.

Despite these limitations, we successfully detected the motor protein MyoH and observed a clear reduction in its signal upon ASAF1 depletion. These data are now included in the revised manuscript (Supplementary Fig. 13j). We have also added a Western blot showing depletion of ASAF1 following rapamycin induction to further support the efficiency of the knockout.

- Figure 6E: In the final schematic, the author should define what does 'apical rings' is referring to. If I'm not mistaken it should refer to the apical polar ring.

Thank you for pointing this out. The term "apical rings" in the scheme refers to the apical polar ring (APR). We apologize for the ambiguity. We now revised it to APR to avoid confusion and ensure consistency with terminology used throughout the manuscript.

- page 13 line 29: 'huge' is quite subjective.

Now we changed it to 'high molecular weight'. (Page 15 line 14)

- Page 14 line 17: Could the author specify the data that supports the claim that 'contact of mature PCRs with the plasma membrane is required for its/their stability'? Despite being an appealing hypothesis, this claim seems to not be supported by any data presented here.

We thank the reviewer for pointing this out. We agree that the manuscript did not present direct experimental evidence to support the claim that contact with the plasma membrane is required for PCR stability. To avoid overstating our interpretation, we have removed this sentence from the revised manuscript. We appreciate the reviewer's attention to this and have ensured that speculative ideas are now clearly framed as such throughout the text.

- Page 14 line 21: The rhoptry positioning data must be reinforced by additional experiments before drawing such conclusions.

We now toned down our statement in manuscript and also discussed this in our discussion section. (Page 16, line 4-19)

“The absence of docked rhoptries in some CGP depleted parasites in our tomogram suggest that CGP might play a role in stabilizing apical complex structures involved in rhoptry positioning, such as AV. However, we never observed a parasite lacking both AVs and rhoptries, indicating that AV loss alone does not account for the absence of rhoptries in our tomograms. One possible explanation is technical: the thin and elongated rhoptry necks may be poorly resolved in thicker tomographic samples that were not subjected to cryo-FIB milling. Supporting this, we observed two instances in wild-type parasites where docked rhoptries were not visible, suggesting this may be a general limitation of the method. In the future, it may be interesting to employ other high-resolution imaging techniques, such as focused ion beam scanning electron microscopy (FIB-SEM), to accurately assess rhoptry formation and apical positioning. This could help address whether PCRs play a role in rhoptry biogenesis or anchoring rhoptries. If the absence of PCR also has a significant effect on rhoptry positioning at the apical tip, similar to defects seen after ICMAP deletion, which affect the ICMTs, this may suggest a link between the ICMTs and PCRs. However, we were unable to observe a direct effect on ICMTs in absence of the PCRs, confirming that ICMT formation doesn’t depend on PCRs.”

- Page 15 line 4: Could the author perhaps clarify why they believe Pcr5 plays a lesser role in PCR stability compared to CGP or Pcr4?

In the original text, the phrasing “probably Pcr5” was used because we did not directly investigate Pcr5 in this study. However, we acknowledge that previous work has clearly demonstrated that Pcr5 is a critical component of the pre-conoidal rings. To reflect this, we have revised the text to include Pcr5 alongside CGP and Pcr4, acknowledging its established role in PCR stability based on prior studies (see page 16, lines 29–31 and page 17, lines 1-2).

“Based on these data we propose a two-step process for PCR formation: initial assembly requires factors that recruit PCR proteins to the nascent conoid complex, independently of known structural proteins. Later these interactions are stabilised, once the PCRs are fully developed and hatch from the mother cell (Fig. 7). This final maturation and stabilisation of PCRs requires CGP, Pcr4 and Pcr5.”

- Page 15 line 11: In absence of CGP, no PCRs can be seen on mature cells. I think there is no ‘mature PCR’ in absence of CGP.

The reviewer is correct that PCRs are absent in the absence of CGP. What we intended to convey is that in developing daughter cells, PCRs can still be observed, whereas they are absent in non-dividing “mother/mature” cells during the G1 phase. We have clarified this point in the text to avoid confusion. (Page 16, lines 22-28)

- Page 15 line 21: ‘...for the formation of daughter conoid complexes’

Now have changed to ‘ASAF1 is exclusively required for the formation of daughter conoid complex’ for accuracy. (Page 17 line 12).

Reviewer #2 (Remarks to the Author):

The manuscript from Li W, Koczy O et al investigated in the apicomplexan parasite *Toxoplasma gondii* (*T. gondii*) the role of CGP, a protein localized at the pre-conoidal rings (PCR) and previously identified by the lab as essential for parasite motility. Inducible depletion of CGP combined with ultrastructural analysis (using cryo-electron tomography) showed that CGP is essential for stabilisation of the PCRs

and recruitment of the F-actin nucleation protein FRM1 in mature conoid after daughter cell replication.

By using proximity-ligand binding assay with CGP as bait, they identified novel putative conoidal proteins possibly involved in conoid complex assembly. The authors further characterized in detail the function of ASAF1 and found that it is required for the early stage of conoid assembly in nascent daughter cells.

Depletion of ASAF1 resulted in failure of conoid complex assembly, disorganised subpellicular microtubules, and lack of IMC formation resulting in impaired daughter cell budding.

The conclusions drawn by the authors are supported by robust experimental work and convincing data sets. The results obtained are novel and interesting. In particular, the strength of the study lies on the combined use of high resolution (or ultrastructural) microscopy and live imaging. It is this spatio-temporal level that allowed the authors to identify new mechanisms of conoid complex assembly in *T. gondii*.

I have only minor comments.

Figure 1:

Please provide Western blots confirming depletion of CGP upon rapamycin treatment and the consequence on the examined conoidal proteins by IFA. Notably, is total amount of FRM1 reduced in CGP-depleted parasites compared to control parasites ? Or depletion of CGP only impacts on FRM1 localization at the mature conoid ?

Please see also our response to reviewer 1. We thank the reviewer for this valuable suggestion. Unfortunately, due to the large size of CGP (~500 kDa) and its relatively low expression, we were unable to detect the protein by Western blot, even under optimized conditions. This limitation is consistent with prior mass spectrometry datasets, such as the conoid proteome reported by Hu et al. (2006), in which CGP was not detected. However, in our previous publication (Li et al., 2022), we confirmed efficient gene excision in approximately 75% of parasites following DiCre induction, and demonstrated CGP depletion by IFA.

Regarding FRM1 and other large apical proteins such as PCKMT and Dap1, we encountered similar technical challenges in detecting them by Western blot. These difficulties are likely attributable to both low expression levels and their substantial molecular sizes. As a result, we have revised the text to take a more cautious tone:

“While we attempted to quantify FRM1, PCKMT, and Dap1 under the same conditions, the results were inconclusive—likely due to their low abundance and large molecular size. Nonetheless, we cannot exclude the possibility that their total levels remain unchanged or are modestly reduced in the absence of CGP.” (Page10, line 14-17)

In contrast, for smaller and more readily detectable proteins such as Pcr4 and ICAP16, we were able to obtain clear Western blot signal. These data are now included in the revised manuscript (Supplementary Fig. 8d). We have also updated the text to read:

“To assess the expression levels of apical proteins that lose their apical localization upon CGP depletion, we performed Western blot analysis. This revealed a reduction in Pcr4 and ICAP16 levels following induction of the *cgp* knockout (Supplementary Fig. 8d), suggesting that these proteins are degraded in mature PCRs” (Page 10, line 11-14)

Line 9 : « Consequently, the defect in gliding motility upon CGP depletion results from the absence of FRM1-dependent F-actin nucleation at the apical tip in good agreement with previous studies ». Please

modulate the conclusion as the authors did not investigate and demonstrate that F-actin nucleation is impaired in CGP-depleted parasites.

We now changed the text, and it reads now

“Consequently, the defect in gliding motility upon CGP depletion is likely due to the absence of FRM1 at the apical tip, where it may fail to mediate F-actin nucleation, consistent with previous studies” (Page 5, line 19-21)

Figure 2

STED-acquired signal for CGP does not display the expected resolution from STED, probably due to weak expression of this protein. Could the authors address the co-localization between PCR4 and CGP using expansion microscopy as they have done for FRM1 to confirm the localization of CGP to the PCRs.

We thank the reviewer for this thoughtful comment. As noted in our general response regarding the use of STED versus Expansion Microscopy (ExM), the apical proteins analyzed in this study—such as CGP and PCR4—are often expressed at relatively low levels, which presents technical challenges for achieving optimal high-resolution imaging of both proteins simultaneously. In practice, this often results in one protein displaying weaker signal or reduced resolution, whether using STED or ExM.

In previous work, we demonstrated CGP localization near the apical complex through co-localization with markers such as SAS6L and RNG2. In this manuscript, we provide further confirmation by showing co-localization with PCR4 via STED microscopy (Fig. 1d), which aligns with and complements our earlier findings.

We believe that the current data sufficiently support CGP’s localization to the PCRs and clearly convey the intended conclusion to the reader.

Could the authors comment on the observation that upon CGP depletion, FRM1 is still recruited to conoids assembling in daughter cells in contrast to the conoid of the mother cell? In both structures, CGP is absent. This suggests that CGP is not required for FRM1 recruitment but only for mature PCR stability. Please comment this point in the discussion.

Now we added this point in the discussion section.

“Surprisingly, even the association of PCR proteins, like FRM1, Dap1, or PCKMT is not affected during conoid formation in replicating parasites. Instead, FRM1 is present during early, mid, and late budding stages until daughter cells hatch from the mother. This suggests that CGP is exclusively required for maintaining the stability of the PCRs on mature parasites, but not during their formation.” (Page 16, line 23-28)

Fig 2E : as designed, the graph is difficult to read. Please modify it by indicating the percentage of CGP-positive conoids in mother versus nascent daughter parasites in both conditions : without and with rapamycin treatment. The authors may for example choose to examine only parasites in M phase (that include both mature and assembling conoids) to perform this quantification.

We have now updated the figure (see Fig. 1h) to distinguish between parasites in replicating and non-replicating stages, based on our IFA observations, and categorized them into four groups. Schematic diagrams have been included to enhance readability.

Line 19: « We speculate that this stabilisation step requires the contact of PCR proteins with the plasma membrane which only occurs after hatching of the daughter cells from the mother. »

Indeed, this hypothesis could likely explain why the PCRs are no longer present in the mature apical conoid of the mother cell.

Did the authors identify in their TurboID assay CGP partners that could contribute to PCR anchoring at the plasma membrane?

See also our response to reviewer 1. At this stage we do not have experimental data supporting this hypothesis. Based on the localization of CGP, we identified some preconoidal proteins in the TurboID assay, such as PCKMT, Dap1, ICAP16. However, depletion of PCKMT does not affect PCR formation supported by CryoET (Qin et al., manuscript in preparation). Similarly, depletion of Dap1 does not impact PCR protein localization, indicating that it is unlikely to contribute to potential PCR anchoring at the plasma membrane. To test this hypothesis further investigation is required, which is outside the scope of this study.

Figure 3

« However, in 33% of *cgp* iKO parasites, rhoptries could not be observed within the tomogram field of view (Fig. 3b and Extended Data Fig. 30 Fig. 2; Supplementary Video 4) »

Please clarify : the absence of rhoptries within the conoid is a consequence of impaired anchoring at the apical tip of the parasite or is there a general defect in rhoptry formation ?

Consequently, please provide immunofluorescence images (expansion microscopy would be best) showing rhoptry formation and localization at different steps of parasite cell cycle in CGP-depleted parasites and provide quantification if any defect is observed.

Currently, we do not fully understand why rhoptries are not observed in 33% of CGP iKO parasites in our tomograms. In our previous study (Li et al., 2022), we assessed rhoptry biogenesis using immunofluorescence assays with markers such as ROP1 and ROP2–4, and we did not detect any obvious defects in rhoptry formation upon CGP depletion.

In the present study, we observed defects in rhoptry secretion in CGP-depleted parasites. Additionally, we identified a loss of the apical vesicle (AV)—a structure known to dock the rhoptry neck—in a subset of parasites. Specifically, IFA analysis using the AV marker Nd6 and tomographic reconstructions showed that ~20% of CGP-depleted parasites lacked an AV at the apical tip. Importantly, we never observed a case where both the AV and rhoptries were simultaneously absent. This suggests that the absence of the AV is not directly responsible for the lack of rhoptries in the tomogram field of view. Additionally, ICMTs, which are reported to dock the rhoptry, were not affected in CGP depleted parasites. This indicates the missing rhoptry is not because of ICMTs.

One possible explanation for the absence of visible rhoptries is technical: due to the thin and elongated nature of rhoptry necks, they can be poorly resolved in some sections of the reconstructed tomograms in thick samples that were not subjected to cryo-FIB-milling. Supporting this interpretation, we observed two instances in wild-type parasites where no docked rhoptries were apparent in the tomograms, indicating that this may be a general limitation of the method.

We appreciate the reviewer's suggestion to use expansion microscopy to further examine rhoptry formation and positioning. However, based on the findings of Dos Santos Pacheco et al. (2021), ExM is not ideally suited to resolving the apical tip of the rhoptry neck, limiting its utility for this purpose. In future work, other high-resolution techniques, such as focused ion beam scanning electron microscopy (FIB-SEM), may help clarify whether PCRs play a role in rhoptry formation or anchoring. We have now included this point in the Discussion section. (Page 16, line 3-18)

Extended Data Fig. 3 / Table 1

Biotin labelling shows a specific staining in what seems to be rhoptries or Golgi? can the author clarify this point and comment.

Endogenous biotinylated proteins are present in *Toxoplasma* in the apicoplast and, to a lesser extent, in the mitochondria. This likely explains why wild-type parasites (Δ Ku80), even without FRM1 or CGP tagged with TurboID, still show biotin signal in these regions.

Please provide a Venn diagram showing the number and identity of common versus unique proteins identified as partners of CGP versus FRM1 by TurboID.

We have now included the requested information in Supplementary Figure 4c.

The authors successfully tagged 13 candidate partners of CGP. Are these candidates common to FRM1 and CGP or unique to CGP?

We thank the reviewer for the opportunity to clarify this point. Of the 13 successfully tagged candidate proteins, 10 were identified as common hits in both the CGP and FRM1 TurboID datasets. One candidate was unique to CGP, and two were unique to FRM1. This information has now been clearly indicated in Supplementary Table 1 for the reader's reference.

Extended Data Fig 4

Many of the selected candidates are not localized at the apical tip of parasites or nascent conoids except for 293480 and 212780 (the latest being only present in nascent conoids). Could the authors comment on this important point and include a comment in the manuscript?

We appreciate the reviewer's comment and agree that only two of our selected candidates—TGGT1_293480 and TGGT1_212780—showed clear localization to the apical tip or nascent conoids. This result is not entirely unexpected given the nature of the BioID approach, which relies on proximity-based biotinylation rather than direct physical interaction. As such, BioID can label proteins that are in close spatial or temporal proximity to the bait, including transiently associated or neighbouring proteins that may not be stable components of the complex. It is also important to note that *T. gondii* expresses several endogenously biotinylated proteins, which may contribute to background signal in mass spectrometry datasets.

Our candidate selection strategy included proteins with either known apical localization based on published studies, predicted apical localization with a final HyperLOPIT probability score of 1, or no assigned localization. This intentionally broad approach was designed to identify both confirmed and potentially novel apical proteins, though it inevitably resulted in candidates that did not localize to the apical complex upon validation.

We have now added a clarifying statement to the Discussion section to reflect these considerations (page 17, lines 3–10).

Extended Data Fig. 6h

Please quantify the plaque assay (also for extended Fig 9e)

All the plaque assays are now quantified.

As previously mentioned for FRM1, please investigate by WB whether absence of Dap1 and PCKMT signals at the mature conoid in CGP-depleted parasites is linked to their (at least partial) degradation ? or only mis-localization as written by the authors. If these proteins are degraded (knowing that dividing parasites with CGP signal in daughter cell conoids represent only about 20% of the total

population) then replace the term « mis-localization » as the protein is rather no longer present than found at another localization within the parasite.

As mentioned above, we made several attempts to perform western blotting on the affected apical proteins, specifically those that lose their apical signal upon CGP depletion, as examined in this study. Unfortunately, only Pcr4 and ICAP16 yielded successful results; other proteins, such as DAP1 and PCKMT, were undetectable, likely due to their low expression levels and relatively large molecular sizes. Our western blot analysis revealed decreased levels of Pcr4 and ICAP16 following CGP knockout induction (Supplementary Fig. 8d), suggesting that these proteins are indeed degraded to some extent in mature parasites. To reflect this accurately, we now avoid using the term “mislocalise” and instead refer to the “absence” or “loss” of these proteins.

Discussion

Can the authors comment on why ASAF1 depletion lead to the absence of IMC formation in nascent daughter cells? Is that an indirect effect linked to the loss of apical polarity in nascent daughter cells or do ASAF1 could play a more direct role on IMC formation?

We have now added a paragraph in the Discussion section addressing this point (page 18, lines 11–15).

“ASAF1 depletion also results in the absence of typical IMC formation in replicating cells. Instead, remnants of potentially nascent IMC structures (Fig. 4a,c) can be observed in the cytosol of the parasites. This indicates that initiation of IMC biogenesis still occurs; however, in the absence of apical polarity and the characteristic SPMT basket, which normally supports alveolar plate architecture²⁰, regular IMC plates fail to form. “

At this stage, we consider the absence of IMC formation in ASAF1-depleted parasites to be a downstream consequence of disrupted apical polarity and defective conoid complex assembly. However, we cannot exclude the possibility that ASAF1 might also play a more direct role in IMC biogenesis. As we currently lack experimental data to distinguish between these possibilities, we have clearly framed this interpretation as speculative and look forward to further studies to explore this potential connection.

Reviewer #3 (Remarks to the Author):

Wei Li et al in the manuscript “Novel apical ring protein essential for conoid complex assembly and daughter cell formation in *Toxoplasma gondii*” have applied a variety of methods with a focus on imaging and showed that that previously identified conoid gliding protein is crucial for targeting actin nucleator formin 1 and other preconoidal ring proteins to preconoidal rings of *Toxoplasma gondii*. They applied cryo-correlative light and electron microscopy as well as cryo-electron tomography to study conoid gliding protein KO parasites, which convincingly revealed an absence of rhoptries (preconoidal ring). Overall, the manuscript is mainly dedicated to the parasitology community, however, the cryo-CLEM method applied is very useful and can be adapted to a variety of projects. It is well-written and presents high-quality imaging data using state-of-the-art imaging techniques. Several minor issues should be addressed in particular to provide statistics on cryo-ET data and to clarify and better present the data for a broader audience. I have mainly focused on assessing the cryo-EM part of the manuscript.

1) Please clarify and consolidate the nomenclature and terms used throughout of the manuscript. For example, rhoptries are not indicated in the legend of Figure 3b and it is not clear if preconoidal ring and rhoptries refer to the same thing.

During the revision process we standardized the nomenclature we use to refer to the *Toxoplasma* cellular structures. On the Figure 2b the segmentation depicts preconoidal rings, conoid, apical polar

ring, microtubules, intra-conoidal microtubules, membrane and inner membrane complex. To keep the clarity of the segmentation, we did not segment all of the cellular structures that can be observed in the parasites, however these structures can be seen on some of the slices of the tomograms on the Figure 2c (such as micronemes, rhoptries and microtubule associated vesicles). Preconoidal rings are two concentric ring-shaped structures located near the parasite's apex, while rhoptries are club-shaped secretory organelles with thin necks and bulbous bottom parts and span from within the conoid to around $\frac{1}{2}$ of the parasite cell length. Rhoptries are indicated on the Figure 3b legend in the line 3.

2) While tomogram data are of high quality, only overview slices are shown. It would be beneficial to show zoom-in of the rhoptries. Also, the color code chosen in the segmentation and segmentation legend (Figure 3a) are not easy to follow as some of the colors are similar. Also indicate that each row represents the same tomogram, perhaps use labelling – bottom slice, central slice, top slice of the tomogram.

We would like to thank the reviewer for pointing out the suggestions on improving Figure 3 (now is Figure 2a-b). We implemented them in the revised version of our manuscript. The colour palette chosen for highlighting the cellular structures on the tomograms as well as for the segmentation was selected to ensure maximal interpretability for colour vision deficiency individuals.

3) Line 13, line 38, page 7: Please provide the exact numbers of tomograms which were analyzed and report the frequency of the observed events:

- “While analysing the cryo-ET data, we noticed that the apical plasma membrane was often disrupted during this process.”

Toxoplasma parasites undergo mechanical stress during vitrification on the cryo-EM grids. During backside blotting of the excess media from around the parasites on the grid, the force created by rapid movement of the liquid towards the paper through the holes of the EM-grid can flatten the parasites. As a result, a portion of parasites becomes damaged, having disrupted membranes or being completely disintegrated, some parasites become slightly flattened and some are unaffected. A precise quantification of the percentage of the parasites with damaged membranes is not possible due to the fact that intact, round parasites are not-electron transparent, especially if clustered together.

For the analysis of the preconoidal rings absence and overall condition of the other organelles we only considered parasites with intact membranes around the apical part of the parasite to eliminate the chance of accounting the damaged parasites as any biological phenotype.

- “However, in 33% of cgp iKO parasites, rhoptries could not be observed within the tomogram field of view (Fig. 3b and Extended Data Fig. 2; Supplementary Video 4).”

During the process of revising our manuscript we analyzed 15 additional parasites. Overall, we analyzed 38 parasites from two separate biological experiments and found that 26.09% (6 parasites out of 23) cgp iKO parasites did not have the rhoptries docked in the conoid in the observed field of view. For the WT parasites, 13.33% (2 out of 15 analyzed) were lacking the rhoptries docked in the observed field of view.

4) Do all wild-type parasites contain preconoidal ring? Does the absence of preconoidal ring has any influence of overall shape and size of the conoid?

Yes, all wild-type parasites contain the preconoidal rings.

We did not observe any changes in the overall shape and size of the conoid when the preconoidal rings are missing. The space in the conoid where the rings would be present remains empty. We however

did not perform any in-depth analysis of the changes in size as we believe it would not be informative due to the likely flattening of the parasites in the vitrification process, as outlined above in more detail.

5) It would be beneficial to introduce and cite previous cryo-ET studies on Toxoplasma apical ring. For example: <https://doi.org/10.1073/pnas.2111661119>; <https://doi.org/10.1038/s41467-023-37327-w>

We would like to thank the reviewer for pointing out the missing references. We included them in the revised version of our manuscript. (Page 7, line 30)

6) Throughout the manuscript, cryo-EM/ET abbreviations could be better used. Sometimes authors use cryoET, **cryo-ET**, cryo-electron microscopy instead of cryo-EM etc.

We would like to thank the reviewer for pointing out the inconsistent abbreviations. We corrected them in the revised version of our manuscript.

7) Methods Title:

“Ultrastructural TEM” should be changed to Thin-section TEM of Epoxy-embedded samples

Changed.

“Cryo-correlated light and electron microscopy images alignment” should be changed to “Correlation of cryo-light and electron microscopy data”

We would like to thank the reviewer for the suggestions. We implemented them in the revised version of our manuscript.

8) Cryo-ET data acquisition (method section): “fluence” is not commonly used and it should be replaced by electron dose rate in e-/Angstrom²/sec. Total dose should be reported in e-/Angstrom².

We thank the reviewer for the suggestion and pointing out the correct units we overlooked. We acknowledge that while “fluence” is not a commonly used term it is the correct one. We added both terms to the Methods section.

9) Please provide info if any motion correction procedure was applied during tomogram reconstruction.

Initially the tomograms were not motion corrected as no subsequent subtomogram averaging was planned to be performed. During the past months however our lab implemented warp2 pipeline and all tomograms were re-reconstructed using this pipeline, with warp2 motion correction.

10) What was used as correlation markers for cryo-CLEM, please specify the features that were used, this might not be obvious to a broad audience.

We thank the reviewer for this comment. We implemented the suggested changes in the Methods section.

Reviewer #4 (Remarks to the Author):

In this paper, Li et al, study the role of a previously identified protein CGP in conoid formation and find that it is important for the stability of the pre-conoidal rings in mature conoids. Using proximity-based labelling approaches, they then identify 15 other candidates that are likely involved in the formation of the conoid and go on to characterise the most promising candidate, which they term ASAF1. This

protein appears to be involved in early conoid assembly and hence SPMT localisation and daughter cell formation.

This is a nice story and will be of interest to the parasite community. The conoid and SPMTs are an exciting topic at the moment with many groups studying them and this paper brings us closer to understanding how these important and complex structures are assembled, organised and stabilised. The data is solid, clear and uses an impressive array of imaging techniques that result in beautiful figures. Some parts could be improved by providing more numerical data (eg. how many cells imaged and of those how many had X phenotype) and the wording of some speculations could be toned down in places.

I only have minor comments:

Abstract and Introduction

- Line 1 page 2: “The conoid complex is a unique structure in apicomplexan parasites”

Is the conoid unique to apicomplexan parasites? I think it is also found in some other alveolates.

We thank the reviewer for this insightful comment. Indeed, the conoid is not exclusive to apicomplexan parasites and has also been identified in other alveolates, including colpodellids, perkinsids, and chromerids. To avoid confusion and maintain consistency with the scope of our study, we have revised the relevant sections of the abstract and introduction to focus specifically on *T. gondii*. See also response to reviewer 1.

- Line 3 and 4 page 2: “It is a dynamic organelle that plays an important role for initiation of gliding motility” and Line 14 and 15 page 3 “From the APRs the minus end of 22 subpellicular MTs radiate and extend to approximately two-thirds of the parasite's length”

The abstract and introduction are written as if this is about all apicomplexa – however the details are *Toxoplasma* specific. Eg. Most *Plasmodium* motile stages don't have a conoid and they don't have 22 microtubules. Either make the abstract and intro just about *Toxoplasma* or make it more general and then remove specific statements like these (and add references for other apicomplexa).

As mentioned above, we now make the introduction *Toxoplasma* specific.

Results and Discussion:

- Lines 28 and 29 on page 7: “However, in 33% of cgp iKO parasites, rhoptries could not be observed within the tomogram field of view”.

Can you clarify this – so in 67% you can see proper localisation of rhoptries? And in what percentage of WT can you see rhoptries within the tomogram field of view?

If you see proper localisation in 67% of iKO cells and not in 100% of WT cell (for example) then I don't think the lines 2 and 3 on pg 6 “our findings highlight the importance of PCRs in ensuring the proper attachment of rhoptries to the apical complex.” can be stated.

Please also add how many tomograms were collected and analysed of WT and iKO to give us a better understanding of the significance of 33%.

See also response to reviewer 1 and 2. We thank the reviewer for the comment and the suggestion for better clarification. During the process of revising our manuscript we analyzed 15 additional parasites. Overall, we analyzed 38 parasites from two separate biological experiments and found that 26.09% (6 parasites out of 23) cgp iKO parasites did not have the rhoptries docked in the conoid in the observed

field of view. For the WT parasites, 13.33% (2 out of 15 analyzed) were lacking the rhoptries docked in the observed field of view. Now we have toned it down to

“Additionally, our findings highlight the potential importance of PCRs in ensuring the proper attachment of rhoptries to the apical complex.” (Page 8, lines 22-23)

and also discussed this in the discussion section (Page 16, line 4-19).

- Line 18 page 11: “Subsequent analyses in regular imaging”

What is “regular imaging”?

By “regular imaging,” we intended to refer to standard IFA imaging performed without expansion microscopy. To improve clarity, we have revised the text to read: “Subsequent analyses by standard (non-expanded) IFA imaging...” (Page 12, lines 20-26)

- For the TEM analysis page 11 – please give numbers – how many WT and ASAF1 KO cells were imaged, for both how many cells did you see daughter conoids or correct IMC formation. Please also have a supplemental figure showing more cells than the one for each WT and KO in figure 5b.

During our TEM analysis, we examined samples from three independent biological replicates. In total, over 100 vacuoles were screened for both wild-type (WT) and ASAF1-inducible knockout (iKO) parasites, although not all were imaged at high resolution. Our analysis focused on vacuoles undergoing daughter cell formation.

In the WT samples, 100% of vacuoles displayed typical architecture with well-formed tachyzoites. Among these, 20–30% showed clear signs of daughter cell formation, and approximately one-third of those allowed unambiguous visualization of daughter conoids. In contrast, in the ASAF1-depleted parasites, we did not observe any cases of proper daughter cell formation or identifiable conoid structures beyond early division stages, likely due to the severe disruption of internal organization.

We have now included these details in the main text (page 13, lines 3–11) and added a new supplementary figure to illustrate representative TEM images from both conditions (Supplementary Fig. 11).

- Lines 16 and 17 on page 14: “This appeared especially important, since our analysis suggests that contact of mature PCRs with the plasma membrane is required for its stability.” I don’t understand where you got this from – what analysis have you done to suggest this? This isn’t clear to me and should be removed (unless you have the analysis to show this).

See our response to reviewer 1 and 2. At this stage we do not have experimental data supporting this hypothesis. Based on the localization of CGP, we identified some pre-conoidal proteins in the TurboID assay, such as PCKMT, Dap1, ICAP16. However, depletion of PCKMT does not affect PCR formation supported by CryoET (Qin et al., unpublished data). Similarly, depletion of Dap1 does not impact PCR protein localization, indicating that it is unlikely to contribute to potential PCR anchoring at the plasma membrane. To test this hypothesis further investigation is required, which is outside the scope of this study.

- Line 22 on page 14 – ICMTs is used for the first time and not defined.

“ICMTs” (intraconoidal microtubules) appears for the first time on the page 3 line 8-9 and the abbreviation is explained there.

- Lines 2,3,4 on page 15: “We therefore speculate that contact of the PCRs to the plasma membrane is required for final maturation and stabilisation of the PCRs, which then requires CGP, Pcr4 (and probably Pcr5).”

Again, as above, I don't understand where this speculation has come from. Here it is called a speculation and above it is called “our analysis”

See above. We removed this speculation.

- Figure 4b. Why does the scale change in images 1-6. It would make it much easier to understand what was going on if this was kept consistent.

Now we have unified the scale and updated with better images.

- Figure 6e. The paper sometimes feels like multiple stories and what I am missing is something that links all the proteins analysed in this paper together in a clear way. I would at least include CGP in this model figure.

Now, we have included CGP in the model.

- Some of the discussion reads like a repeat of the results, this could be made more succinct.

We have revised the Discussion section to reduce repetitions.

Materials and methods:

- Ultrastructural TEM – missing a few pieces of information like pixel size.

We updated the Material and Methods and defined the pixel size.

- Tomogram reconstruction – again missing information, what software was used for frame alignment (assuming you collected frames – this is not mentioned in cryo-electron tomography), CTF correction etc.

Initially the tomograms were not motion corrected as no subsequent subtomogram averaging was planned to be performed. During the past months however our lab implemented warp2 pipeline and all tomograms were re-reconstructed using this pipeline, with warp2 motion correction and CTF correction and these tomograms are included in the revised figures, except for the Supplementary Videos 2 and 3, where not-motion corrected tomograms were used.

References:

- Koreny, L., M. Zeeshan, K. Barylyuk, E.C. Tromer, J.J.E. van Hooff, D. Brady, H. Ke, S. Chelaghma, D.J.P. Ferguson, L. Eme, R. Tewari, and R.F. Waller. 2021. Molecular characterization of the conoid complex in *Toxoplasma* reveals its conservation in all apicomplexans, including *Plasmodium* species. *PLoS Biol.* 19:e3001081.
- Li, W., J. Grech, J.F. Stortz, M. Gow, J. Periz, M. Meissner, and E. Jimenez-Ruiz. 2022. A splitCas9 phenotypic screen in *Toxoplasma gondii* identifies proteins involved in host cell egress and invasion. *Nat Microbiol.* 7:882-895.